

# Effects of meteorology and emissions on urban air quality: a quantitative statistical approach to long-term records (1999–2016) in Seoul, South Korea

Jihoon Seo[1,2], Doo-Sun R. Park[3], Jin Young Kim[1], Daeok Youn[4], Yong Bin Lim[1], Yumi Kim[5]

[1]Green City Technology Institute, Korea Institute of Science and Technology, Seoul, 02792, South Korea
[2]School of Earth and Environmental Sciences, Seoul National University, Seoul 08826, South Korea
[3]Department of Earth Sciences, Chosun University, Gwangju 61452, South Korea
[4]Department of Earth Science Education, Chungbuk National University, Cheongju 28644, South Korea
[5]Division of Resource and Energy Assessment, Korea Environment Institute, Sejong 30147, South Korea

*Correspondence to*: Jin Young Kim (jykim@kist.re.kr), Daeok Youn (dyoun@chungbuk.ac.kr)

**Abstract.** Together with emissions of air pollutants and precursors, meteorological conditions play important roles in local air quality through accumulation or ventilation, regional transport, and atmospheric chemistry. In this study, we extensively investigated multi-timescale meteorological effects on the urban air pollution using the long-term measurements data of $PM_{10}$, $SO_2$, $NO_2$, CO, and $O_3$ and meteorological variables over the period of 1999–2016 in Seoul, South Korea. The long-term air quality data were decomposed into trend-free short-term components and long-term trends by the Kolmogorov-Zurbenko filter, and the effects of meteorology and emissions were quantitatively isolated using a multiple linear regression with meteorological variables. In terms of short-term variability, intercorrelations among the pollutants and meteorological variables and composite analysis of synoptic meteorological fields exhibited that the warm and stagnant conditions in the migratory high-pressure system are related to the high $PM_{10}$ and primary pollutant while the strong irradiance and low $NO_2$ by high winds at the rear of cyclone are related to the high $O_3$. In terms of long-term trends, decrease in $PM_{10}$ ($-1.75\ \mu g\ m^{-3}\ yr^{-1}$) and increase in $O_3$ ($+0.88\ ppb\ yr^{-1}$) in Seoul were largely contributed by the meteorology-related trends ($-0.94\ \mu g\ m^{-3}\ yr^{-1}$ for $PM_{10}$ and $+0.47\ ppb\ yr^{-1}$ for $O_3$), which were attributable to the subregional scale wind speeds increase. Comparisons with estimated local emissions and socioeconomic indices like GDP growth and fuel consumptions indicate probable influences of the 2008 global economic recession as well as the enforced regulations from the mid-2000s on the emission-related trends of $PM_{10}$ and other primary pollutants. Change rates of local emissions and transport term of long-term components calculated by the tracer continuity equation revealed a decrease in contributions of local emissions to the primary pollutants including $PM_{10}$ and an increase in contributions of local secondary productions to $O_3$. This study shows meteorological conditions related to the episodic air pollution events in short-term and provides insights into the current environmental policies and regulations by isolation of emission-related trends in long-term.



# 1 Introduction

Over the past few decades, rapid urbanization, population and economic growth, and increase in energy consumption have exacerbated air pollution in the developing countries of South and East Asia (Sun et al., 2016; Liu et al., 2017; Shi et al., 2018). In response to growing concerns about air quality deterioration, several East Asian countries have implemented strict regulations on emissions in recent decades and achieved some degree of improvement in the primary air pollutants. For example, ambient concentrations of carbon monoxide (CO), sulfur dioxide ($SO_2$), and nitrogen oxides ($NO_x$) have decreased after the 1990s in South Korea and after mid-2000s in China owing to various efforts to reduce their emissions (e.g., desulfurization on coal-fired power plants and industries, installation of selective catalytic reduction equipment, use of low-sulfur fuels and natural gas, and enhancement of vehicle emission standards; Shon and Kim, 2011; Klimont et al., 2013; Ray and Kim, 2014; van der A et al., 2017; C. Li et al., 2017; Kim and Lee, 2018). Despite the efforts to reduce the primary pollutants and secondary precursors, however, the East Asian countries still suffer from frequent severe haze pollutions (Huang et al., 2014) and experience continuous increasing of ozone ($O_3$) levels (Seo et al., 2014; Sun et al., 2016).

Such a discrepancy between the past reduction of emissions and the current continuous air pollution can be minimized by considering effects of meteorological conditions on the air quality. The meteorological conditions often play important roles in local air quality through accumulation or ventilation of pollutants, regional transport of polluted or clean air, and atmospheric chemistry for the formation of secondary species (Zheng et al., 2015; Seo et al., 2017). For example, during the Chinese severe haze episode in January 2013, active secondary aerosol formation in the stagnant surface conditions and shallow boundary layer over the North China Plain induced the extremely high particulate matter (PM) concentrations without abrupt changes in emissions (Huang et al., 2014; Zheng et al., 2015; Cai et al., 2017; Zou et al., 2017). Meteorologically adjusted long-term PM trend in Beijing shows that unfavorable meteorological condition has reduced efficiency of recent emission control (Zhang et al., 2018). Although the meteorological effects on the current increase of $O_3$ in China are unclear (Ma et al., 2016), the $O_3$ pollution is expected to become a more important issue in the future warmer climate (Jacob and Winner, 2009; Lin et al., 2008; Schnell et al., 2016) due to its high dependence on temperature (Sillman and Samson, 1995; Lin et al., 2001).

Considering the important role of meteorological conditions on the local air pollution levels, long-term urban air quality data must be interpreted carefully with examining both meteorological effects and changes in local and regional emissions. The meteorological conditions contribute to the various timescale changes in urban air quality by synoptic scale weather in short-term (Seo et al., 2017) and climatic variabilities in long-term (Cai et al., 2017; Zou et al., 2017; Oh et al., 2018), as well as its inherent seasonality (Kim et al., 2018). On the other hand, the changes in emissions mostly occur in long-term timescale due to the implementation of policy and regulations (van der A et al., 2017) and economic boom or recession (Vrekoussis et al., 2013; Tong et al., 2016). Thus, temporal decomposition of air pollution measurement data into different timescales provides useful information on the meteorological effects on both day-to-day fluctuations, seasonal characteristics, and long-term trend of the air pollution levels. In addition, the effectiveness of the past regulations on emissions and the influences of



socioeconomic changes can be more reasonably assessed by considering and isolating the meteorological effects on air pollution.

As one of the highly populated megacities in East Asia, the Seoul metropolitan area (SMA), which has a population of 25 million and 9 million vehicles, is an apparent large emission source of various air pollutant species (NIER, 2016). To mitigate

air pollution in the SMA by regulation on primary emissions, the South Korean government enacted the Special Act on the Improvement of Air Quality in Seoul Metropolitan Area in 2003 and enforced the act in 2005 (Kim and Lee, 2018). In fact, long-term measurements in Seoul shows that concentrations of $PM_{10}$ (PM with aerodynamic diameter $\leq 10$ $\mu m$) and other primary pollutants have been decreased over the decades, and these long-term decreasing trends have been regarded as results from the enhanced environmental regulations (Ghim et al., 2015; Kim and Lee, 2018). However, considering that the air quality

of Seoul is largely affected by the transport of regional air pollutants and the synoptic meteorological conditions (Seo et al., 2017), the efficiency of the current emission control policy should be evaluated after understanding and quantitative isolating of the meteorological effects on the long-term measurement data. Using regional air quality simulation, Kim et al. (2017) recently reported that interannual variability in winds could fluctuate the $PM_{10}$ levels in Seoul despite the emission control efforts. However, quantification of the meteorological effects based on a direct comparison between measured air pollutants

and meteorological variables are further needed to demonstrate the effects of the emission controls.

In this study, we aimed to extensively investigate the role of meteorological conditions on the air pollution in Seoul, South Korea, and quantitatively examine the contributions of meteorology and emissions using the statistical approach to 18-yr (1999–2016) long records of ground $PM_{10}$, $NO_2$, $SO_2$, CO, and $O_3$ together, as well as various local meteorological variables. To decompose the air pollution time series into different timescales and isolate the meteorological effects on the air pollution,

we employed a simple statistical concept using the Kolmogorov-Zurbenko (KZ) filter and multiple linear regression model with meteorological factors, which has been used in previous studies (Wise and Comrie, 2005; Seo et al., 2014; Henneman et al., 2015; Ma et al., 2016; P. Li et al., 2017; Zhang et al., 2018). Synoptic meteorological conditions related to episodic air pollution events were investigated using composite analyses of the trend-free short-term components, and Isolation of the meteorology- and emission-related components from the long-term air pollution trends were conducted. We further explored

the long-term changes in contributions of local emissions and transport to the air pollution in Seoul using a simplified tracer continuity equation. Finally, the identified emission-related trends of each pollutant species were compared with the estimated emission inventories, as well as socioeconomic indices that could affect local emissions.

## 2 Data

The Korean Ministry of Environment has measured $PM_{10}$ and four gaseous air pollutants of $SO_2$, nitrogen dioxide ($NO_2$), CO,

and $O_3$ at 264 urban air quality monitoring sites in South Korea and provides their 1 h average concentrations (NIER, 2017). Daily average concentrations for $PM_{10}$, $SO_2$, $NO_2$, and CO and daily maximum 8-h average concentration for $O_3$ ($O_{3\ 8h}$) at each site were derived from the hourly data. We choose 18 sites located in Seoul to obtain daily air pollutant concentrations





representative for air quality of Seoul (Figs. 1a and S1a), based on data availability of more than 90% for 1999–2016. Although the selected 18 sites are spread over Seoul, concentrations from individual sites show fair spatial homogeneity (e.g., coefficient of variations for $PM_{10}$ at the selected sites are ~0.17; Fig. S1b and c). Since Seoul is located in a basin surrounded by mountainous terrain (higher than ~500 m above sea level) except its western exit of the Han River toward the Yellow Sea, air

pollutants from both emissions and transport can be stagnated and mixed in the basin in the prevailing westerlies.

The most noticeable features in the air pollutants data in Seoul are decreasing of $PM_{10}$ and increasing of $O_3$ levels in the long-term timescale. Annual average $PM_{10}$ concentrations and exceedance days of the South Korean air quality standard (AQS; 100 $\mu g\ m^{-3}$) were decreased from 69 $\mu g\ m^{-3}$ to 41 $\mu g\ m^{-3}$ and from 64 days to 2 days between 1999 and 2015 (Fig. 1b). Note that we excluded episodic Asian dust (AD) days (183 days for the analysis period) from the $PM_{10}$ analysis to focus on the

anthropogenic sources, although AD did not much affect the long-term $PM_{10}$ trend. On the other hand, annual average $O_{3\ 8h}$ levels and exceedance days of the AQS (60 ppbv) have been increased from 25 ppbv to 39 ppb and from 5 days to 58 days between 2002 and 2016, respectively (Fig. 1c).

In terms of meteorological characteristics in Seoul, we used daily averages of temperature (°C), sea level pressure (hPa), relative humidity (%), wind speed (m s$^{-1}$), and solar irradiance (W m$^{-2}$) at the Seoul weather station (Fig. 1a) managed by the

Korea Meteorological Administration. Note that the air quality monitoring sites in Seoul are spread over the area within a radius of ~15 km from the weather station, and such spatial size is enough to resolve the influence of synoptic conditions on the local meteorological factors. To investigate the synoptic meteorological conditions, the geopotential height and wind fields at 850 hPa, 10 m wind fields, total cloud cover, and surface solar radiation were derived from the European Centre for Medium-Range Weather Forecasts Reanalysis Interim (ERA-Interim) data.

We additionally employed estimated local emission data and several socioeconomic indices that could affect the emissions to compare with the separated emission-related air pollution trends. The annual estimated emissions of sulfur oxides (SO$_x$), NO$_x$, CO, volatile organic compounds (VOCs), and $PM_{10}$ in Seoul from the Clean Air Policy Support System (CAPSS) inventory (Lee et al., 2011; NIER, 2016), the national gross domestic product (GDP) growth (IMF, 2017) and the annual consumptions of the final energy, petroleum, and anthracite in Seoul (KEEI, 2016) were used in this study.

## 3 Decomposition of air pollutant time series

### 3.1 Temporal decomposition of air pollutant time series

Time series of daily air pollutant concentration at a given place comprises mainly three components: trend, seasonality, and white noise (Rao and Zurbenko, 1994). The trend (long-term component) is attributable to the long-term variations in local and regional emissions related to socioeconomic status and policies (Mijling et al., 2013) and the long-term changes in

meteorological conditions, which can affect atmospheric chemistry and regional transport patterns (Cai et al, 2017; Zou et al., 2017). The seasonality (seasonal component) arises from the seasonal variations of meteorological conditions (Kim et al., 2018) and energy consumption pattern (Zhu et al., 2013). The white noise (short-term component) is a day-to-day variation mainly



related to synoptic-scale weather changes, which control accumulation/ventilation of local air pollutants and transport of regional air pollutants (Seo et al., 2017) and is partly associated with short-term fluctuations in local emissions (Russell et al, 2010).

To decompose the air pollutant time series into these three components in different timescales, we used the KZ filter that has

been employed in many air pollution time series studies, particularly on $O_3$ and PM (Wise and Comrie, 2005; Seo et al., 2014; Ma et al., 2016; P. Li et al., 2017). The KZ filter, here denoted as $KZ_{(m,p)}$, represents $p$ times iteration of moving average of time width $m$ (days) and removes high-frequency component, of which period is smaller than the effective filter width, $N$ ($\geq m \times p^{1/2}$). The KZ filter method is applicable to the time series with missing data owing to iterative moving average process and provides high accuracy level comparable to that of the wavelet transform method, albeit its simplicity (Eskridge et al.,

1997). Here we applied $KZ_{(15,5)}$ and $KZ_{(365,3)}$ filters, which can remove variabilities of the periods shorter than 33 days and 1.7 yr, to filter out the short-term component and to leave the long-term component, respectively. The long-term component can be further separated into the emission-related trend and the meteorology-related trend by isolation of the emission-related component using a multiple linear regression model with representative meteorological variables.

Note that the original concentration ($\chi$) was transformed into its natural logarithm values ($X = \ln\chi$) prior to the decomposition.

Because the number distribution of daily air pollutant concentrations is usually lognormal (e.g., Fig. S2), the natural log-transformation of the original concentration data is required for proper temporal decomposition with the KZ filter (Eskridge et al., 1997). The detailed decomposition procedure is described in followings and schematically summarized with $PM_{10}$ time series in Seoul in Fig. 2.

The time series of log-scaled pollutant concentration can be expressed as the sum of short-term ($X_{ST}$), seasonal ($X_{SN}$), and

long-term ($X_{LT}$) components.

$$X(t) = X_{ST}(t) + X_{SN}(t) + X_{LT}(t) \tag{1}$$

The sum of seasonal and long-term components is a baseline ($X_{BL} = X_{SN} + X_{LT}$). $X_{BL}$ and $X_{ST}$ can be easily decomposed by applying the $KZ_{(15,5)}$ filter to $X$, which filters out the white noise-like $X_{ST}$, as follows:

$$X_{BL}(t) = KZ_{(15,5)}X(t) = X(t) - X_{ST}(t) \tag{2}$$

Now the baseline can be assumed to consist of its repeated climatological seasonal cycle ($X_{BL}^{clm}$) and residuals ($\varepsilon$).

$$X_{BL}(t) = X_{BL}^{clm}(t) + \varepsilon(t) \tag{3}$$

Although $X_{BL}^{clm}$ obviously occupies most of the seasonality in $X_{BL}$, $\varepsilon$ also contains some minor seasonal variability unconsidered in $X_{BL}^{clm}$ together with the long-term trend. To obtain the long-term component ($X_{LT}$) by filtering out the minor seasonality, the $KZ_{(365,3)}$ filter is applied to $\varepsilon$ as follows:

$$X_{LT}(t) = KZ_{(365,3)}\varepsilon(t) = X_{BL}(t) - X_{SN}(t) \tag{4}$$





Then the seasonal component ($X_{\mathrm{SN}}$), which represents the sum of the pure seasonal climatology ($X_{\mathrm{BL}}^{\mathrm{clm}}$) and the minor seasonality ($\varepsilon - \mathrm{KZ}_{(365,3)}\varepsilon$), can be obtained by difference between $X_{\mathrm{BL}}$ and $X_{\mathrm{LT}}$.

Note that if we define $\chi_{\mathrm{BL}} = \exp(X_{\mathrm{BL}})$ and $\chi_{\mathrm{ST}} = \exp(X_{\mathrm{ST}})$ and employ the similar concept to the relationship between the original concentration and its log-transformation ($\chi = \exp(X)$), $\chi_{\mathrm{BL}}$ represents the baseline concentration of the air pollutant,

and $\chi_{\mathrm{ST}}'$ becomes the ratio of original concentration to baseline concentration ($\chi/\chi_{\mathrm{BL}}$). Similarly, $\exp(X_{\mathrm{SN}})$ and $\exp(X_{\mathrm{LT}})$ can be defined as $\chi_{\mathrm{SN}}$ and $\chi_{\mathrm{LT}}$, respectively. Then $\chi_{\mathrm{SN}}$ represents the seasonal change in concentration without trend, and $\chi_{\mathrm{LT}}$ becomes the ratio of baseline concentration to detrended seasonal concentration ($\chi_{\mathrm{BL}}/\chi_{\mathrm{SN}}$).

As shown in the example with PM$_{10}$ in Seoul, the KZ$_{(15,5)}$ filter effectively removes PM$_{10_{\mathrm{ST}}}$, of which period is smaller than 33 days, and leaves both the seasonality of high PM$_{10}$ concentrations in winter and spring and the long-term decreasing trend

in PM$_{10_{\mathrm{BL}}}$ (Figs. 2 and S4b). PM$_{10_{\mathrm{SN}}}$ and PM$_{10_{\mathrm{LT}}}$ well represent the seasonal variation, of which periods are between 33 days and 1.7 yr with representative periodicities of 0.5 yr and 1 yr, and the long-term variations, of which period are longer than 1.7 yr, respectively (Figs. 2 and S4c–d). The high levels in winter and spring in PM$_{10_{\mathrm{SN}}}$ in Seoul is attributable to the shallow boundary layer that traps local pollutants near the ground and frequent regional transport from China during the cold season (Kim et al., 2018).

**3.2 Separation of emission- and meteorology-related trends**

Since the long-term variability in air pollutant concentrations can be affected not only by changes in local and regional emissions but also by changes in meteorological conditions, the long-term trend is assumed to be consisted of meteorologically-adjusted (emission-related) long-term component ($X_{\mathrm{LT}}^{\mathrm{emis}}$) and meteorology-related long-term component ($X_{\mathrm{LT}}^{\mathrm{met}}$). Therefore, the baseline can be represented as follows:

$$X_{\mathrm{BL}}(t) = X_{\mathrm{SN}}(t) + X_{\mathrm{LT}}^{\mathrm{met}}(t) + X_{\mathrm{LT}}^{\mathrm{emis}}(t) \qquad (5)$$

To isolate the term $X_{\mathrm{LT}}^{\mathrm{emis}}$ in Eq. (5), we built a multiple linear regression model employing the baseline time series of the five representative meteorological variables (MET$_{\mathrm{BL}}$), such as temperature (T$_{\mathrm{BL}}$), sea level pressure (P$_{\mathrm{BL}}$), relative humidity (RH$_{\mathrm{BL}}$), wind speed (WS$_{\mathrm{BL}}$), and solar irradiance (SI$_{\mathrm{BL}}$), which are obtained by the KZ$_{(15,5)}$ filter, as follows:

$$X_{\mathrm{BL}}(t) = a_0 + \sum_i a_i \mathrm{MET}_{\mathrm{BL}_i}(t) + \varepsilon'(t),$$

$$\mathrm{MET}_{\mathrm{BL}} = [\mathrm{T}_{\mathrm{BL}}, \mathrm{P}_{\mathrm{BL}}, \mathrm{RH}_{\mathrm{BL}}, \mathrm{WS}_{\mathrm{BL}}, \mathrm{SI}_{\mathrm{BL}}] \qquad (6)$$

where $\varepsilon'$ is a sum of the non-meteorological long-term variability ($X_{\mathrm{LT}}^{\mathrm{emis}}$) and the minor seasonal variability unexplained by the multiple linear regression model ($\varepsilon' - X_{\mathrm{LT}}^{\mathrm{emis}}$). Note that daily maximum temperature (T$_{\mathrm{max}_{\mathrm{BL}}}$) was used for the model of O$_{3\,8h_{\mathrm{BL}}}$ instead of daily average temperature (T$_{\mathrm{BL}}$) considering daytime O$_3$ as in previous study (Seo et al., 2014). By removing the minor seasonality from $\varepsilon'$ using the KZ$_{(365,3)}$ filter, $X_{\mathrm{LT}}^{\mathrm{emis}}$ can be isolated as follows:





$$X_{\mathrm{LT}}^{\mathrm{emis}}(t) = \mathrm{KZ}_{(365,3)}\varepsilon'(t) = X_{\mathrm{LT}}(t) - X_{\mathrm{LT}}^{\mathrm{met}}(t) \tag{7}$$

Then $X_{\mathrm{LT}}^{\mathrm{met}}$ can be simply obtained by difference between $X_{\mathrm{LT}}$ and $X_{\mathrm{LT}}^{\mathrm{emis}}$.

In the example displayed in Fig. 2, $\mathrm{PM_{10}}_{\mathrm{LT}}$ in Seoul show a continuous decrease between 2003 and 2012. Such a decreasing

trend in $\mathrm{PM_{10}}$ has been recognized as a result of the reduction in diesel vehicle emissions and fugitive dust in Seoul (Ghim et

al., 2015; Kim and Lee, 2018). However, a recent modeling study suggested that the long-term increase in wind speed might

additionally contribute to the past improvement of $\mathrm{PM_{10}}$ air quality in Seoul (Kim et al., 2017). In fact, both $\mathrm{PM_{10}}_{\mathrm{LT}}^{\mathrm{emis}}$ and

$\mathrm{PM_{10}}_{\mathrm{LT}}^{\mathrm{met}}$ in Fig. 2 show decreasing patterns, and this supports probable influences of both emission controls and meteorology

on the $\mathrm{PM_{10}}$ trend in Seoul.

**3.3 Contributions of local emissions and transport to the long-term trends**

$X_{\mathrm{LT}}^{\mathrm{emis}}$ contains both changes in local emissions ($X_{\mathrm{LT}}^{\mathrm{emis(L)}}$) and changes in transport of regional emissions ($X_{\mathrm{LT}}^{\mathrm{emis(T)}}$), and thus

$X_{\mathrm{LT}}$ can be represented as Eq. (8).

$$X_{\mathrm{LT}}(t) = X_{\mathrm{LT}}^{\mathrm{emis(T)}}(t) + X_{\mathrm{LT}}^{\mathrm{emis(L)}}(t) + X_{\mathrm{LT}}^{\mathrm{met}}(t) \tag{8}$$

Then the rate of change of $X_{\mathrm{LT}}$ should satisfy a simple continuity equation as follows:

$$\frac{\partial X_{\mathrm{LT}}}{\partial t} = -\vec{V}_{\mathrm{LT}} \cdot \nabla X_{\mathrm{LT}} + S_{\mathrm{LT}} \tag{9}$$

where the advection term, $-\vec{V}_{\mathrm{LT}} \cdot \nabla X_{\mathrm{LT}}$, represents the transport of regional emissions mainly by horizontal winds, and the

sources and sinks term, $S_{\mathrm{LT}}$, can be regarded as the long-term changes by both local emissions and chemical production,

accumulation, and dissipation by meteorological factors. Because $X_{\mathrm{LT}}$ and $X_{\mathrm{LT}}^{\mathrm{met}}$ were already known, the rates of change of

$X_{\mathrm{LT}}^{\mathrm{emis(T)}}$ and $X_{\mathrm{LT}}^{\mathrm{emis(L)}}$ can be derived as follows:

$$\frac{\partial}{\partial t}\left(X_{\mathrm{LT}}^{\mathrm{emis(T)}}\right) = -\vec{V}_{\mathrm{LT}} \cdot \nabla X_{\mathrm{LT}} = -u_{\mathrm{LT}}\left(\frac{\partial X_{\mathrm{BL}}}{\partial x} - \frac{\partial X_{\mathrm{SN}}}{\partial x}\right) - v_{\mathrm{LT}}\left(\frac{\partial X_{\mathrm{BL}}}{\partial y} - \frac{\partial X_{\mathrm{SN}}}{\partial y}\right) \tag{10}$$

$$\frac{\partial}{\partial t}\left(X_{\mathrm{LT}}^{\mathrm{emis(L)}}\right) = \frac{\partial X_{\mathrm{LT}}}{\partial t} - \left[\frac{\partial X_{\mathrm{LT}}^{\mathrm{met}}}{\partial t} + \frac{\partial}{\partial t}\left(X_{\mathrm{LT}}^{\mathrm{emis(T)}}\right)\right] \tag{11}$$

where $\vec{V}_{\mathrm{LT}} = (u_{\mathrm{LT}}, v_{\mathrm{LT}})$ is long-term zonal and meridional winds components representative for the target area, and $\nabla X_{\mathrm{LT}}$ is

horizontal gradient of long-term component of air pollutant for the larger area, which are identical to difference between $\nabla X_{\mathrm{BL}}$

and $\nabla X_{\mathrm{SN}}$. If $\nabla X_{\mathrm{LT}} > 0$ at specific time, therefore, the horizontal gradient of baseline ($\nabla X_{\mathrm{BL}}$) at that time is steeper than that of

seasonal climatology ($\nabla X_{\mathrm{SN}}$).

In this study, we choose 70 air quality monitoring sites over the SMA, which are spread over the area within a radius of 50 km

from the Seoul weather station, based on data availability of more than 75% for 1999–2016 (Fig. 1a). Daily $X_{\mathrm{BL}}$ and $X_{\mathrm{SN}}$ at

those sites were utilized to determine $\nabla X_{\mathrm{BL}}$ and $\nabla X_{\mathrm{SN}}$ by linear regressions of $X_{\mathrm{BL}}$ and $X_{\mathrm{SN}}$ on the zonal and meridional



distances from the weather station and finally to obtain $\nabla X_{LT}$ ($= \nabla X_{BL} - \nabla X_{SN}$; Figs. S5 and S6c, e, g, i, and k). Also, $u_{LT}$ and $v_{LT}$ in Seoul is calculated by the $KZ_{(365,3)}$ filter with wind direction and speed at the Seoul weather station (Fig. S6a and b). From $\vec{V}_{LT}$ and $\nabla X_{LT}$ of each species, we calculated the long-term transport terms for each air pollutant species in Seoul (Figs. S6d, f, h, j, and l).

In Eq. (11), $\frac{\partial X_{LT}}{\partial t}$ and $\frac{\partial X_{LT}^{met}}{\partial t}$ are three orders of magnitude smaller than $\frac{\partial}{\partial t}\left(X_{LT}^{emis(L)}\right)$ and $\frac{\partial}{\partial t}\left(X_{LT}^{emis(T)}\right)$, and thus, approximately $\frac{\partial}{\partial t}\left(X_{LT}^{emis(L)}\right) \approx -\frac{\partial}{\partial t}\left(X_{LT}^{emis(T)}\right)$. For example, if we assume that $\chi \sim 50\ \mu g\ m^{-3}$, $\frac{\partial \chi}{\partial t} \sim -10\ \mu g\ m^{-3}$ decade$^{-1}$, $\frac{\partial \chi}{\partial x} \sim -10\ \mu g\ m^{-3}$ (100 km)$^{-1}$, and $u \sim 1\ m\ s^{-1}$ at a given place, of which conditions are similar to PM$_{10}$ in Seoul and its metropolitan area, the transport term $-u\frac{\partial X}{\partial x}$ ($= -\frac{u}{\chi}\frac{\partial \chi}{\partial x}$) $\sim -2 \times 10^{-6}\ s^{-1}$, while the tendency term $\frac{\partial X}{\partial t}$ ($= \frac{1}{\chi}\frac{\partial \chi}{\partial t}$) $\sim -2.5 \times 10^{-9}\ s^{-1}$. If we further assume that the local PM$_{10}$ emission in Seoul (area of 605 km$^2$) has the same order of magnitude as the transport term in concentration

($u\frac{\partial \chi}{\partial x} \sim 1 \times 10^{-4}\ \mu g\ m^{-3}\ s^{-1}$) and the boundary layer height over the area is about 1 km, the total amount of yearly PM$_{10}$ emission in Seoul will be approximately 1.9 kt, which is consistent with estimation by the CAPSS emission inventory (Fig. 5a; NIER, 2016). In fact, because air pollutant concentrations are close to the steady state for the long-term period, its tendency is a small residual between two large terms: influx of clean or polluted air (transport) and local emissions/production or dissipation/deposition (sources and sinks). It should be noted that the transport term and the source/sink term, however, are

not balanced each other in short-term timescale because the day-to-day rate of change of the air pollution concentration is comparable to both transport and source/sink terms.

## 4 Application to air pollutants in Seoul

Variances of each timescale component of PM$_{10}$, SO$_2$, NO$_2$, and CO in Seoul were generally largest in the short-term fluctuation (~50–70%), while that of O$_{3\ 8h}$ were largest in its seasonal cycle (Table 1). This indicates an important role of

synoptic-scale weather on the day-to-day variations of the primary air pollutants concentrations in Seoul. The changes in synoptic flow pattern control local air quality by accumulation/ventilation and regional transport of air pollutants (Zheng et al., 2015; Seo et al., 2017). On the other hand, O$_3$ is a photochemical product and thus is controlled largely by the annual cycle of solar irradiance in such a NO$_x$- and VOCs-rich urban area, although its short-term variability, which is caused by the changes in synoptic weather and related precursor concentrations and irradiance, is comparable to the seasonal variability.

In terms of the long-term trends, variabilities related to the long-term local and regional emission changes ($X_{LT}^{emis}$) are comparable to those induced by the long-term meteorological effects ($X_{LT}^{met}$) for PM$_{10}$, NO$_2$, CO, and O$_{3\ 8h}$, while $X_{LT}^{emis}$ is dominant for total long-term variabilities of SO$_2$ (Table 1). Since the local-scale meteorology could affect more on the local emission-related long-term trend rather than on the regional background-related long-term trend, the smaller variability of $X_{LT}^{met}$ imply the less influence of the local emission changes compared to the regional background level changes on $X_{LT}^{emis}$.



## 4.1 Meteorological effects on seasonality in air pollution

In the process of isolating $X_{LT}^{emis}$ from $X_{LT}$, the multiple linear regression model predicting $X_{BL}$ from meteorological predictors ($MET_{BL}$) were used. The model using the meteorological variables explains well $O_{3\,8h_{BL}}$ (adjusted $R^2 = 0.86$), while the baseline of $PM_{10}$ ($PM_{10_{BL}}$) is much less explained by the model (adjusted $R^2 = 0.51$; Table 2). This indicates that the long-term non-meteorological (or emission-related) variability in $PM_{10_{BL}}$ is much larger than that in $O_{3\,8h_{BL}}$. In the linear regression model, $PM_{10_{BL}}$ and the baselines of primary gaseous pollutants of $SO_2$, $NO_2$, and CO commonly show strong positive correlation with $P_{BL}$ but strong negative correlations with $T_{BL}$ and $RH_{BL}$, while $O_{3\,8h_{BL}}$ shows strong positive correlations ($p < 0.05$) with $SI_{BL}$ and $T_{max_{BL}}$ but strong negative correlation ($p < 0.05$) with $P_{BL}$ (Table 2). Meteorological conditions over the Korean Peninsula in winter is affected by a cold and dry Siberian High that induces both the shallow boundary layer to trap the primary air pollutants near the surface and the northwesterly winds to transport the regional pollutants from the continent (Kim et al., 2018) and thus elevates $PM_{10_{BL}}$ in winter to early spring (Fig. S3b). On the other hand, enhanced photochemistry by strong irradiance together with temperature effects on $O_3$ formation (increasing of biogenic hydrocarbons and hydroxyl radicals and enhanced thermal decomposition of peroxyacetyl nitrate in warm condition; e.g. Sillman and Samson, 1995) elevates $O_{3\,8h}$ levels in late spring and summer in Seoul (Fig. S3b).

## 4.2 Synoptic influences on short-term air pollution events

The short-term air pollution variability is closely related to the day-to-day variations of local meteorological factors. Intercorrelations among $X_{ST}$ of air pollutant species and meteorological variables in Seoul are summarized in Table 3. Note that using $X_{ST}$ instead of the original daily concentrations has an advantage to provide short-term features unbiased to seasonal characteristics and background levels because $\exp(X_{ST})$ is equivalent to the ratio of measured concentration to filtered baseline concentration as aforementioned in Sect. 3.1.

Significant positive intercorrelations ($p < 0.01$) among $PM_{10_{ST}}$ and primary gaseous pollutants ($SO_{2_{ST}}$, $NO_{2_{ST}}$, and $CO_{ST}$) together with their strong negative relationships to $T_{ST}$ and $WS_{ST}$ ($p < 0.01$) indicate that the high $PM_{10}$ episodes in Seoul were occurred in warm and stagnant weather conditions. Such warm and stagnant conditions can be induced by the migratory high-pressure system (e.g., negative relationships of $WS_{ST}$ to $T_{ST}$ and $P_{ST}$ in Table 3) or by the warm front of the extratropical cyclone (e.g., negative correlations of $T_{ST}$ to $WS_{ST}$ and $P_{ST}$ but its positive correlation to $RH_{ST}$ in Table 3). The lag composites of the geopotential height and wind anomalies at 850 hPa (about 1.5 km of altitude) relative to each day of $\exp(PM_{10_{ST}})$, that is equivalent to the ratios of measured concentration to the filtered baseline concentration of $PM_{10}$, exceeding the sum of its mean and one standard deviation ($> 1.50$) indicate that the high $PM_{10}$ episodes in Seoul is associated with the former condition (Fig. 3). As shown in Fig. 3a–c, a high-pressure anomaly develops over southern China and moves eastward in three days before the high $PM_{10}$ days. This kind of synoptic pattern can induce both slow regional transport of secondary precursors from China along the northern boundary of the high-pressure system (Fig. 3b and c) as well as gradual accumulation of primary and



secondary aerosols in local area in the high-pressure system (Fig. 3d; Seo et al., 2017). Strong correlations of $PM_{10_{ST}}$ with $SO_{2_{ST}}$ and $NO_{2_{ST}}$ ($p < 0.01$; Table 3) imply possible large contributions of the secondary species such as sulfate and nitrate aerosols to the haze episodes in Seoul. In fact, previous studies reported that the secondary inorganic aerosols proportion to the fine mode PM ($PM_{2.5}$) was measured as high as ~50% in average and even reached to ~75% during the severe haze event

(Seo et al., 2017; Kim et al., 2018).

In terms of short-term variability in $O_{3\,8h}$ levels in Seoul, $O_{3\,8h_{ST}}$ are strongly correlated positively with $SI_{ST}$ and $T_{max_{ST}}$ but negatively with $RH_{ST}$ ($p < 0.01$; Table 3). This likely reflects favorable meteorological environment for $O_3$ formation: strong irradiance with warm and dry conditions. Interestingly, $O_{3\,8h_{ST}}$ shows positive correlation with $WS_{ST}$, although stagnant condition is regarded as conducive to $O_3$ formation in general. This is related to the strong negative relationship between

$O_{3\,8h_{ST}}$ and $NO_{2_{ST}}$ and that between $WS_{ST}$ and $NO_{2_{ST}}$. Since the reaction with NO, which is produced by photolysis of $NO_2$, is a major sink of $O_3$, high $NO_2$ concentration reduces $O_3$ on high irradiance days, while the low $NO_2$ concentration can increase $O_3$. In addition, $NO_{2_{ST}}$ and $CO_{ST}$ are strongly correlated ($r = +0.86$) because both CO and $NO_x$ are mainly emitted by transportation exhaust in Seoul (82% of CO and 67% of $NO_x$ emissions; NIER, 2016), and thus $O_{3\,8h_{ST}}$ has negative relationship with $CO_{ST}$, as like that with $NO_{2_{ST}}$. The composites of the 850 hPa geopotential height and wind anomalies for

high $O_{3\,8h}$ days that $\exp(O_{3\,8h_{ST}})$ exceeds the sum of its mean and one standard deviation ($> 1.41$) shows that the high $O_{3\,8h}$ event frequently occurs at the transition of weather from the cyclonic anomaly to the anticyclonic anomaly (Fig. 4a). As the anomalous low-pressure system moves eastward out of the Korean peninsula, the decrease in cloud cover (Fig. 4b) results in the increase in surface irradiance (Fig. 4c) and thus the $O_3$ level. Since this kind of anomalous synoptic pattern enhances the mean westerly flow by the anomalous northwesterly (Figs. 4a and S7a), the wind speed in this region is increased. As shown

by the strong positive correlations between $WS_{ST}$ and $NO_{2_{ST}}$ (Table 3), $NO_2$ can be reduced by the high winds and thus could contribute to the high $O_3$ concentration in short-term timescale.

## 4.3 Long-term trends of air pollution in Seoul

Times series of $X_{LT}$ and its two decomposed components, $X_{LT}^{emis}$ and $X_{LT}^{met}$, for each air pollutant species are shown in Fig. 5, and their linear trends are summarized in Table 4. Note that these time series range from July 2000 to June 2015 because 546

days at the beginning and ending of data were lost due to truncation effect of the $KZ_{(365,3)}$ filter. The linear trend of $X_{LT}$ represents a fractional change rate (% $yr^{-1}$) of the baseline concentration ($\chi_{BL}$) because $\frac{\partial X_{LT}}{\partial t}$ is equivalent to $\frac{1}{\chi_{BL}}\frac{\partial \chi_{BL}}{\partial t}$. The fractional change rate can be converted into an equivalent concentration change rate by multiplying with time-mean $\chi_{BL}$ for the analysis period.

In Seoul, there are significant decreasing trends ($p < 0.05$) in $PM_{10_{LT}}$ ($-3.6\%\ yr^{-1}$) and $CO_{LT}$ ($-2.9\%\ yr^{-1}$) and an increasing

trend ($p < 0.1$) in $O_{3\,8h_{LT}}$ ($+3.1\%\ yr^{-1}$) for the recent fifteen years, while $NO_{2_{LT}}$ ($-1.4\%\ yr^{-1}$) and $SO_{2_{LT}}$ ($+0.8\%\ yr^{-1}$) doesn't show statistically significant trends (Table 4). The decrease of $PM_{10}$ concentration and increase of $O_{3\,8h}$ level in their long-





term linear trends can be clearly identified in temporal variation of $X_{LT}$ (Fig. 3a and e) and are consistent with the temporal characteristics in their annual average concentrations (Fig. 1b and c). In terms of CO, abrupt decrease in the early 2000s, which was reported to be associated with introduction of natural gas vehicle supply and improvement of fuel quality (Kim and Shon, 2011), affects such a strong linear trend (Fig. 5b). On the other hand, $SO_2$ concentration in Seoul has already been stabilized

at ~5 ppb during the recent decade (Fig. 5c), although it decreased significantly in 1980s and 1990s owing to the government's efforts to control the $SO_x$ emissions by use of low-sulfur fuel and natural gas for industry and transportation (Ray and Kim, 2014; NIER, 2017). $NO_2$ concentration has been varied between 30 and 40 ppb in the recent decade and also shows relatively weak trend (Fig. 5d; NIER, 2017). Interestingly, $NO_2$ level has been stabilized and $NO_x$ level has been even decreased for the period despite increasing of the number of vehicle in Seoul from 2.3 million in 1999 to 3.1 million in 2016, probably owing to

implementation of natural gas vehicles and low emission diesel engines (Shon and Kim, 2011; Kim and Lee, 2018). One explanation for the less decreasing of $NO_2$ compared to NO is the extension of diesel particulate filter (DPF) usage for diesel vehicles that could additionally convert NO to $NO_2$ in the exhaust line (Alvarez et al, 2008; Kim and Lee, 2018). Since Seoul is a $NO_x$-saturated regime area (Jin et al., 2012), such a decreasing trend in ambient $NO_x$ concentration in recent decades may be one of the causes of the long-term increasing of $O_{3\ 8h}$ in Seoul.

Although the decreasing trends in $PM_{10_{LT}}$, $CO_{LT}$, and $NO_{2_{LT}}$ in the recent decade have been regarded as the result of efforts to reduce the local emissions (Kim and Lee, 2018), their $X_{LT}^{emis}$ linear trends are less than half of $X_{LT}$ linear trends (Table 4). This indicates the more important role of the long-term changes in local meteorology in the long-term air pollution trends in Seoul.

In contrast to the comparable linear trends of $X_{LT}^{emis}$ and $X_{LT}^{met}$ of $PM_{10}$, CO, and $NO_2$, the linear trend of $SO_{2_{LT}}$ is dominantly

contributed by that of $SO_{2_{LT}}^{emis}$ (Table 4). This suggests the larger influence of the regional background $SO_2$ than that of the local $SO_x$ emission on $SO_{2_{LT}}^{emis}$ because the long-term effects of local meteorological conditions on the local emission-related air pollution trend must be limited if the long-term change in local emission is negligible. In fact, the estimated $SO_x$ emission intensity of Seoul (7.4 t km$^{-2}$) in the reference year 2010 was much lower than that of an industrial port city in the west adjacent to Seoul, Incheon (18.1 t km$^{-2}$), or the Chinese eastern coastal region of Jing-Jin-Ji (Beijing-Tianjin-Hebei), Shandong, Jiangsu,

and Shanghai (14.3 t km$^{-2}$; NIER, 2016; M. Li et al., 2017). In consideration with prevailing westerly in this region (Fig. S7a), the long-term $SO_2$ concentration trend in Seoul can be more affected by the long-term change in regional background level rather than that in local emission.

In the reference year 2010, estimated emission intensities of CO and $NO_x$ in Seoul in 2010 were estimated as 215.3 t km$^{-2}$ and 117.4 t km$^{-2}$, and these are obviously much higher than those in Incheon (44.1 t km$^{-2}$ for CO and 51.3 t km$^{-2}$ for $NO_x$) or the

Chinese eastern coastal region (98.1 t km$^{-2}$ for CO and 17.4 t km$^{-2}$ for $NO_x$; NIER, 2016; M. Li et al., 2017). Therefore, $CO_{LT}^{emis}$ and $NO_{2_{LT}}^{emis}$ in Seoul are probably much more affected by the long-term changes in local emissions rather than the changes in regional background levels.





### 4.3.1 Meteorology-related long-term trends

For the most of species in Seoul except $SO_2$, the larger decreasing trends of $X_{LT}^{met}$ than that of $X_{LT}^{emis}$ have been found (Table 4). For example, equivalent linear trend of $PM_{10\,LT}$ was $-17.5\ \mu g\ m^{-3}$ decade$^{-1}$, while that of $PM_{10\,LT}^{emis}$ was only $-8.1\ \mu g\ m^{-3}$ decade$^{-1}$. Thus, the recent achievement of ambient $PM_{10}$ reduction in Seoul would be less successful without the contribution

by $PM_{10\,LT}^{met}$ trend ($-9.4\ \mu g\ m^{-3}$ decade$^{-1}$; Table 4 and Fig. 5a). In terms of $O_3$, equivalent linear trend of $O_{3\,8h\,LT}$ ($+8.8$ ppb decade$^{-1}$) was also slightly more contributed by $O_{3\,8h\,LT}^{met}$ ($+4.7$ ppb decade$^{-1}$) compared to $O_{3\,8h\,LT}^{emis}$ ($+4.1$ ppb decade$^{-1}$; Table 4 and Fig. 5e). Therefore, the long-term changes in meteorology that helps improving PM air quality may have made worse the $O_3$ air quality.

Since the only meteorological parameter that shows a significant ($p < 0.1$) linear trend in Seoul is wind speed ($+0.57$ m s$^{-1}$

decade$^{-1}$; Table 4 and Fig. 6a), the meteorological effects on the long-term decreases of $PM_{10}$, CO, and $NO_2$ are probably related to the long-term increase of wind speed that could result in enhancing ventilation of the primary pollutants. Such a role of wind speed in recent changes in PM concentrations in Seoul was also indicated by regional air quality model simulations (Kim et al., 2017). Interestingly, the recent increasing of wind speed in Seoul is a subregional scale change rather than a synoptic or larger scale climate variability. Linear trend distribution of observed wind speed over South Korea for 1999–2016

shows a rough pattern of increasing in the inland area and decreasing in the coastal area (Fig. 6b). However, the significant long-term increases of wind speed near Seoul seems to be somewhat limited to the Han River basin. In addition, although linear trend pattern of atmospheric circulation at 850 hPa (~1.5 km of altitude) over the western North Pacific/East Asia region shows weakening of the Aleutian low during the period (Fig. S7a and b), no significant trend can be observed in both 850 hPa wind fields and 10 m wind speeds over and near the Korean Peninsula (Fig. S7b and c).

Since the major increase in wind speed was observed during two periods (2001–2004 and 2010–2011; Fig. 6a), changes in $PM_{10\,LT}^{met}$, $CO_{LT}^{met}$, and $NO_{2\,LT}^{met}$ mostly occurred in these periods. Such an increase in wind speed could result in more ventilation of $NO_x$ and elevation of $O_3$ levels in the $NO_x$-saturated regime. However, $O_{3\,8h\,LT}^{met}$ shows additional variability related to the long-term variation of solar irradiance (e.g., decrease for 2005–2006 and increase for 2007–2009 in both $O_{3\,8h\,LT}^{met}$ and $SI_{LT}$, Figs. 5e and 6a).

### 4.3.2 Emission-related long-term trends

Although $PM_{10\,LT}$ continuously decreased between 2003 and 2012, a major decrease of $PM_{10\,LT}^{emis}$ occurred between 2008 and 2009 (Fig. 5a). Since the decrease of $X_{LT}^{emis}$ during the period can be found in other primary gaseous pollutants (Fig. 5b–d), it could relate to the reduction of local and/or regional primary emissions. For example, the CAPSS emission inventory shows abrupt decreases in estimated $SO_x$, $NO_x$, and $PM_{10}$ emissions in Seoul from 2007 to 2009 (Fig. 7a). One plausible reason of

such reduction of primary emissions during the period can be attributed to enforcement of the Special Act on the Improvement of Air Quality in Seoul Metropolitan Area, which took effect from January 2005. The act aimed to reduce annual average $PM_{10}$



and NO₂ concentrations to meet 40 µg m⁻³ and 22 ppb by the year 2014, by regulations of $NO_x$, $SO_x$, $PM_{10}$, and VOCs emissions mainly from industries and transportations (Ghim et al., 2015; Kim and Lee, 2018).

Socioeconomic status can also be an important factor for emissions and air pollution (Vrekoussis et al, 2013; Tong et al., 2016; Stern and van Dijk, 2017). In terms of local emissions, the rapid reduction of primary emissions in Seoul between 2008 and

2009 coincided with the rapid drops of South Korean GDP growth (Fig. 7b), which were associated with the 2008 global financial crisis triggered by the US subprime mortgage meltdown (Kim et al., 2017). Final energy consumption in Seoul shrank during the recession, primarily by decrease in petroleum products (especially diesel and liquefied petroleum gases) consumption (Fig. 5b) (KEEI, 2016). The decrease in $PM_{10}$, $NO_x$, and $SO_x$ emissions in Seoul in 2007–2009 were attributable to the reductions of diesel consumption and anthracite consumption (Fig. 7a and c). In terms of regional emissions, temporal

variation of $SO_{2\,LT}^{emis}$, which shows the large decrease in 2008–2009 and slight recovery in following brief period (Fig. 5c), corresponds to interannual variation of satellite $SO_2$ observation over China (van der A et al., 2017). The large decrease in 2008–2009 are attributable to slowdown of the Chinese economy, as well as desulfurization program of the Chinese authority (van der A et al., 2017; C. Li et al., 2017).

Temporal variation of $O_{3\,8h\,LT}^{emis}$ shows a general increasing trend through the analysis period (Fig. 5e). Considering $NO_x$-rich

condition in Seoul, this may results from the long-term decrease of $NO_{2\,LT}^{emis}$ (Fig. 5d) and relatively more reduction of $NO_x$ emission compared to that of VOCs emission in the recent decade (Fig. 7a). However the factors that caused its steep increase in 2005, slight decrease in 2010, and following recovery between 2011 and 2012 are somewhat vague, without further information of concentration and temporal variation of VOCs in Seoul.

### 4.3.3 Long-term effects of local emissions vs. transport

The change rates of $X_{LT}^{emis(L)}$ and $X_{LT}^{emis(T)}$ have nearly same magnitude but opposite signs. Since magnitudes of the change rates of $X_{LT}$, $X_{LT}^{met}$, and $X_{LT}^{emis}$ are much smaller compared to those of $X_{LT}^{emis(L)}$ and $X_{LT}^{emis(T)}$ in the long-term timescale, characteristics of their temporal variation is not easy to be explained by using $X_{LT}^{emis(L)}$ and $X_{LT}^{emis(T)}$. However, comparing these two terms could provide some physical and quantitative insights into temporal changes in each contribution of local emissions and transport of regional background emissions to the local air quality, albeit without considering atmospheric

chemistry. For example, $\frac{\partial}{\partial t}\left(X_{LT}^{emis(L)}\right) > 0$ has implications in the long-term increasing contributions of local emissions and, at the same time, flushing out of air pollutants. On the other hand, $\frac{\partial}{\partial t}\left(X_{LT}^{emis(L)}\right) < 0$ shows gradual reduction of local emissions and compensation by increase in transport of regional background emissions.

In Seoul, $\frac{\partial}{\partial t}\left(X_{LT}^{emis(L)}\right) > 0$ for $PM_{10}$ and primary gaseous pollutants (CO, $SO_2$, and $NO_2$) in the early 2000s, but the change rates got close to zero owing to the efforts to reduce the primary emissions (Fig. 5). Unlike CO and $NO_2$, of which emission

intensities are large in Seoul, $SO_2$ shows obvious positive $\frac{\partial}{\partial t}\left(SO_{2\,LT}^{emis(T)}\right)$ in recent years (Fig. 5c), and this indicates the large



effect of regional transport on the variability of $SO_2$ concentrations in Seoul. Similar to $SO_2$, $PM_{10}$ also show recent increase in $PM_{10LT}^{emis(T)}$ with decreasing of $PM_{10LT}^{emis(L)}$ (Fig. 5a). In terms of $O_3$, $O_{3\,8hLT}^{emis(L)}$ has been gradually changed from decreasing phase to the increasing phase, and this implies that the recent increase of $O_3$ concentration in Seoul probably more related to the production by local precursors, rather than the transport of regional background $O_3$.

Interestingly, the change rates of $X_{LT}^{emis(L)}$ reflect some changes in socioeconomic factors. For example, $PM_{10LT}^{emis(L)}$ and $SO_{2LT}^{emis(L)}$ turned into the decreasing phase since 2008 (Fig. 5a and c), corresponding to the aforementioned descriptions in the previous subsection. In addition, the variation of $\frac{\partial}{\partial t}\left(NO_{2LT}^{emis(L)}\right)$ reflects well the changes in diesel consumption, which were not clearly shown in the CAPSS inventory. The change rate of $NO_{2LT}^{emis(L)}$ was negative between 2006 and 2011 but turned into positive since 2012 (Fig. 5d), and these changes correspond to the diesel consumption in Seoul, which was

decreased from 2007 to 2012 but has increased since then due to policy support for the diesel vehicle market to reduce greenhouse gas emissions (Fig. 7c).

## 5 Conclusions

Based on the statistical approach to the long-term air pollutant measurement data from the urban air quality monitoring sites in Seoul, the present study revealed the important role of synoptic weather conditions on the episodic air pollution events and

the meteorological effects on the long-term air pollution trends. Temporal decomposition using the KZ filter technique and the multiple linear regression with meteorological factors allowed separation of the trend-free short-term variability and isolation of the meteorology- and emission-related long-term trends in the daily air pollution data. Simplified continuity equation using the surface wind data in Seoul and air pollutant measurement data from the wider area around Seoul further provided the approximate estimation of the long-term changes in contributions of local emissions and transport to the air pollution in Seoul.

In terms of the short-term variabilities, which occupies the largest portion of the total air pollution variabilities, the high $PM_{10}$ and primary gaseous pollutants concentrations are related to the influence of migratory high-pressure systems, which induces both regional transport and local accumulation of the pollutants in the warm and stagnant conditions. On the other hand, the high $O_3$ episodes are related to the weather transitions at the rear of cyclones, which accompany the $NO_2$ reduction by high winds and the strong solar irradiance.

In terms of the long-term trends, Seoul experienced a decreasing of $PM_{10}$ concentrations ($-17.5$ $\mu$g m$^{-3}$ decade$^{-1}$) and an increase in $O_{3\,8h}$ levels ($+8.8$ ppb decade$^{-1}$) over the period of 1999–2016. Such long-term changes are largely contributed by the meteorology-related trends (e.g., $-9.4$ $\mu$g m$^{-3}$ decade$^{-1}$ for $PM_{10}$ and $+4.7$ ppb decade$^{-1}$ for $O_{3\,8h}$) related to the decadal increase in surface wind speeds ($+0.57$ m s$^{-1}$ decade$^{-1}$). Therefore, the recent improvement of particulate air quality in Seoul was not achieved solely by the emission control policies but was also induced by changes in local meteorological conditions,

especially, wind speeds. Although no evidence of the influence of synoptic or larger scale climate variability can be found, the long-term wind speed increase was also observed in the Han River basin around Seoul and probably enhanced the ventilation



of local primary pollutants and secondary precursors. Since Seoul is the $NO_x$-saturated regime, the long-term decrease in $NO_2$ by the enhanced winds could be a cause for the long-term increase in $O_{3\ 8h}$ levels. Unlike other species, $SO_2$ doesn't show a significant meteorology-related trend because of the small influence of local meteorology on its small local emissions.

The isolated emission-related air pollution trends of $PM_{10}$ and other primary gaseous pollutants commonly showed a major

decrease between 2008 and 2009. Although the enforcement of the Special Act on the Improvement of Air Quality in Seoul Metropolitan Area in 2005 might affect the reduction of local emissions, socioeconomic indices like GDP growth and energy consumptions also indicate probable influence of the 2008 global economic recession on the changes in the emission-related air pollution trends. Contributions of the local emissions and the transport of the regional background air pollutants to the emission-related long-term trends are almost balanced each other because the change rate of air pollution trend is negligibly

small compared to those of the local emissions and the transport in long-term. The changes in the contributions of local emissions to $PM_{10}$ and primary gaseous pollutants turned into decreasing phase since the mid-2000s in Seoul, and those species, especially $PM_{10}$ and $SO_2$, became to be more affected by the regional background levels in recent years. The contribution of local emissions to $NO_2$ has turned into increasing phase in 2010s due to the recent policy support for the diesel vehicle. Also, the recent increasing phase of local contributions to $O_3$ implies an important role of local secondary production in the increasing

trend of $O_3$ in Seoul.

Although the results from the emission-related long-term air pollution trends showed general agreement with estimated emissions and socioeconomic factors, the statistical technique employed in this study overlooked a role of long-term changes in the chemical conditions and reactions on the urban aerosols and $O_3$ (e.g., atmospheric oxidants, aerosol acidity and hygroscopicity, and secondary organic aerosol formation) and its relationship with the long-term changes in relevant

meteorological factors (e.g., atmospheric water, irradiance, and temperature). Nevertheless, this simple concept of considering the emission-related air pollution trends separately from the meteorology-related trends could give insights into the assessment of past regulations on emissions and help the improvement of current environmental policies on urban air quality.

**Data availability**

The hourly data of $PM_{10}$, $SO_2$, $NO_2$, CO, and $O_3$ concentrations over South Korea for 1999–2016 are available in the website

managed by the Korea Environment Corporation (https://www.airkorea.or.kr). The hourly data of temperature, sea level pressure, relative humidity, wind speed, and solar irradiance at the Seoul weather station for the same period can be found in the website of the KMA (https://data.kma.go.kr). The ERA-Interim data can be accessed via the European Centre for Medium-Range Weather Forecasts (ECMWF) data server (http://apps.ecmwf.int/datasets/).





**Acknowledgements**

This research was supported by the Korea Institute of Science and Technology (KIST) and the National Strategic Project-Fine particle of the National Research Foundation of Korea (NRF) funded by the Ministry of Science and ICT (MSIT), the Ministry of Environment (ME), and the Ministry of Health and Welfare (MOHW) (2017M3D8A1090654). Daeok Youn was supported by Basic Science Research Program through the National Research Foundation of Korea (NRF) funded by the Ministry of Education (2015R1D1A3A01020130).

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



**Table 1: Explained variances (%) of short-term components ($X_{ST}$), seasonal components ($X_{SN}$), and long-term components ($X_{LT}$) of PM₁₀, SO₂, NO₂, CO, and O₃ ₈ₕ in Seoul for the period between July 2000 to June 2015 that $X_{LT}$ data is available.**

| Components | PM$_{10}$ | SO$_2$ | NO$_2$ | CO | O$_{3\,8h}$ | Notes |
|:---:|:---:|:---:|:---:|:---:|:---:|:---|
| $X_{ST}$ | 69.9% | 54.1% | 72.4% | 50.3% | 47.4% | Short-term components |
| $X_{SN}$ | 27.6% | 46.8% | 28.7% | 39.7% | 52.2% | Seasonal components |
| $X_{LT}$ | 8.1% | 5.5% | 3.8% | 15.2% | 5.7% | Long-term components |
| $X_{LT}^{emis}$ | 6.4% | 5.4% | 3.0% | 12.4% | 4.3% | Emission-related $X_{LT}$ |
| $X_{LT}^{met}$ | 6.9% | 0.3% | 1.9% | 10.7% | 4.0% | Meteorology-related $X_{LT}$ |





**Table 2: Correlation coefficients ($r$) between baseline time series of pollutants ($PM_{10}$, $SO_2$, $NO_2$, CO, and $O_{3\,8h}$) and meteorological variables (T, $T_{max}$, P, RH, WS, and SI), and adjusted $R^2$ between the baseline time series of pollutants ($X_{BL}$) and multiple linear regression model ($a_0 + \sum_i a_i MET_{BL_i}$).**

|  |  | $PM_{10_{BL}}$ | $SO_{2_{BL}}$ | $NO_{2_{BL}}$ | $CO_{BL}$ | $O_{3\,8h_{BL}}$ |
|---|---|---|---|---|---|---|
| Correlation coefficients ($r$) between $X_{BL}$ and $MET_{BL}$ | $T_{BL}$ | −0.481[a] | −0.785[a] | −0.704[a] | −0.689[a] | - |
|  | $T_{max_{BL}}$ | - | - | - | - | +0.726[a] |
|  | $P_{BL}$ | +0.338[a] | +0.682[a] | +0.666[a] | +0.664[a] | −0.760[a] |
|  | $RH_{BL}$ | −0.480[a] | −0.627[a] | −0.635[a] | −0.383[a] | +0.169 |
|  | $WS_{BL}$ | −0.031 | +0.287 | −0.013 | −0.203 | +0.217 |
|  | $SI_{BL}$ | −0.001 | −0.366[a] | −0.241 | −0.540[a] | +0.894[a] |
| Adjusted $R^2$ for $a_0 + \sum_i a_i MET_{BL_i}$ | | 0.508 | 0.594 | 0.637 | 0.597 | 0.863 |

[a] The correlation is statistically significant at the 95% level or higher ($p < 0.05$).



**Table 3: Correlation coefficients ($r$) among short-term components of each pollutant and meteorological variable in Seoul for the period between February 1999 and November 2016.**

| $X_{ST}$ | $PM_{10_{ST}}$ | $SO_{2_{ST}}$ | $NO_{2_{ST}}$ | $CO_{ST}$ | $O_{3\,8h_{ST}}$ | $T_{ST}$ | $T_{max_{ST}}$ | $P_{ST}$ | $RH_{ST}$ | $WS_{ST}$ |
|---|---|---|---|---|---|---|---|---|---|---|
| $SI_{ST}$ | +0.072 | +0.110$^a$ | −0.075 | −0.101$^a$ | +0.537$^a$ | +0.032 | +0.283$^a$ | +0.336$^a$ | −0.674$^a$ | −0.015 |
| $WS_{ST}$ | −0.342$^a$ | −0.372$^a$ | −0.687$^a$ | −0.578$^a$ | +0.213$^a$ | −0.258$^a$ | −0.315$^a$ | −0.296$^a$ | +0.015 | |
| $RH_{ST}$ | +0.038 | −0.048 | +0.091$^a$ | +0.219$^a$ | −0.385$^a$ | +0.122$^a$ | −0.081$^a$ | −0.446$^a$ | | |
| $P_{ST}$ | +0.048 | +0.106$^a$ | +0.142$^a$ | +0.031 | +0.048 | −0.194$^a$ | −0.053 | | | |
| $T_{max_{ST}}$ | +0.410$^a$ | +0.448$^a$ | +0.508$^a$ | +0.486$^a$ | +0.118$^a$ | +0.924$^a$ | | | | |
| $T_{ST}$ | +0.373$^a$ | +0.391$^a$ | +0.471$^a$ | +0.465$^a$ | −0.012 | | | | | |
| $O_{3\,8h_{ST}}$ | +0.072 | −0.017 | −0.223$^a$ | −0.223$^a$ | | | | | | |
| $CO_{ST}$ | +0.744$^a$ | +0.718$^a$ | +0.859$^a$ | | | | | | | |
| $NO_{2_{ST}}$ | +0.627$^a$ | +0.687$^a$ | | | | | | | | |
| $SO_{2_{ST}}$ | +0.720$^a$ | | | | | | | | | |

$^a$ The correlation is statistically significant at the 99% level or higher ($p < 0.01$).





**Table 4: Linear trends of long-term components of air pollutants and meteorological variables in Seoul for the period between July 2000 and June 2015.**

| Components | Trends | | $p$-values | Components | Trends | | $p$-values |
|---|---|---|---|---|---|---|---|
| | (% yr$^{-1}$) | (per decade) | | | (% yr$^{-1}$) | (per decade) | |
| $PM_{10_{LT}}$ | $-3.63^a$ | $-17.54^a$ ($\mu$g m$^{-3}$) | 0.027 | $O_{3\,8h_{LT}}$ | $+3.09^a$ | $+8.77^a$ (ppb) | 0.062 |
| $PM_{10_{LT}}^{emis}$ | $-1.69$ | $-8.15$ | 0.153 | $O_{3\,8h_{LT}}^{emis}$ | $+1.44$ | $+4.10$ | 0.179 |
| $PM_{10_{LT}}^{met}$ | $-1.95^a$ | $-9.39^a$ | 0.032 | $O_{3\,8h_{LT}}^{met}$ | $+1.65^a$ | $+4.67^a$ | 0.093 |
| $SO_{2_{LT}}$ | $+0.80$ | $+0.42$ (ppb) | 0.320 | Meteorological variables | | | |
| $SO_{2_{LT}}^{emis}$ | $+0.71$ | $+0.37$ | 0.352 | $T_{LT}$ | | $-0.10$ (°C) | 0.778 |
| $SO_{2_{LT}}^{met}$ | $+0.08$ | $+0.04$ | 0.423 | $Tmax_{LT}$ | | $+0.17$ | 0.710 |
| $NO_{2_{LT}}$ | $-1.36$ | $-4.53$ (ppb) | 0.173 | $P_{LT}$ | | $+0.02$ (hPa) | 0.822 |
| $NO_{2_{LT}}^{emis}$ | $-0.56$ | $-1.88$ | 0.326 | $RH_{LT}$ | | $-1.82$ (%) | 0.135 |
| $NO_{2_{LT}}^{met}$ | $-0.80^a$ | $-2.65^a$ | 0.044 | $WS_{LT}$ | | $+0.57^a$ (m s$^{-1}$) | 0.066 |
| $CO_{LT}$ | $-2.91^a$ | $-1.73^a$ (ppb) | 0.032 | $SI_{LT}$ | | $+4.66$ (W m$^{-2}$) | 0.294 |
| $CO_{LT}^{emis}$ | $-1.09$ | $-0.65$ | 0.106 | | | | |
| $CO_{LT}^{met}$ | $-1.82^a$ | $-1.08^a$ | 0.046 | | | | |

[a] The slope of linear trend is statistically significant at the 90% level or higher ($p < 0.1$).





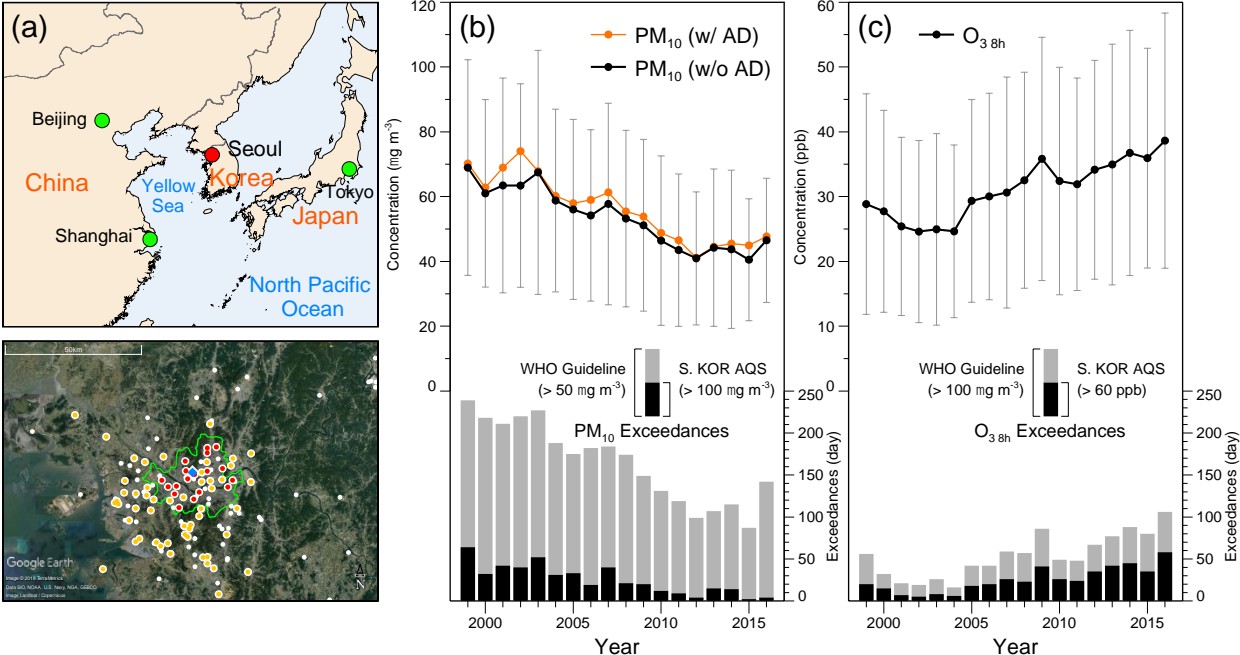

**Figure 1: (a) Geographical locations of air quality monitoring sites in the SMA (circles) and the Seoul weather station (a blue solid diamond). Red solid circles denote 18 air quality monitoring sites utilized to obtain average daily air quality data in Seoul. Yellow solid circles show 52 air quality monitoring sites, which together with 18 sites (red solid circles) were used to calculate spatial gradients of the long-term components of air pollutants. Boundary of Seoul is marked with green line, and the satellite image is a courtesy of Google Earth Pro. Annual average concentrations and exceedances of the World Health Organization (WHO) guidelines and the South Korean AQS for (b) $PM_{10}$ (excluding AD days) and (c) $O_{3\ 8h}$ in Seoul. Annual average $PM_{10}$ concentrations including AD days are additionally represented in orange line in (b). The vertical bars on the annual average concentrations indicate annual standard deviations.**





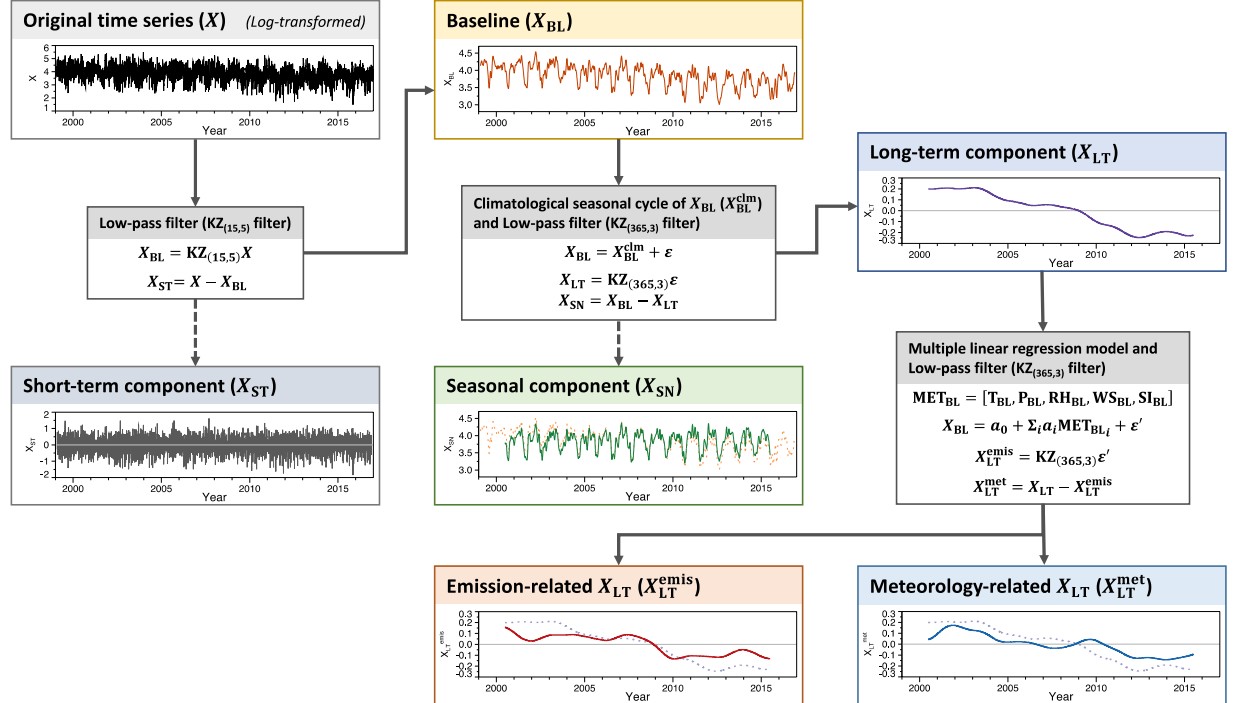

**Figure 2: Schematic flowchart of temporal decomposition of air pollution time series (PM₁₀ in Seoul) into short-term, seasonal, and emission-related and meteorology-related long-term components.**



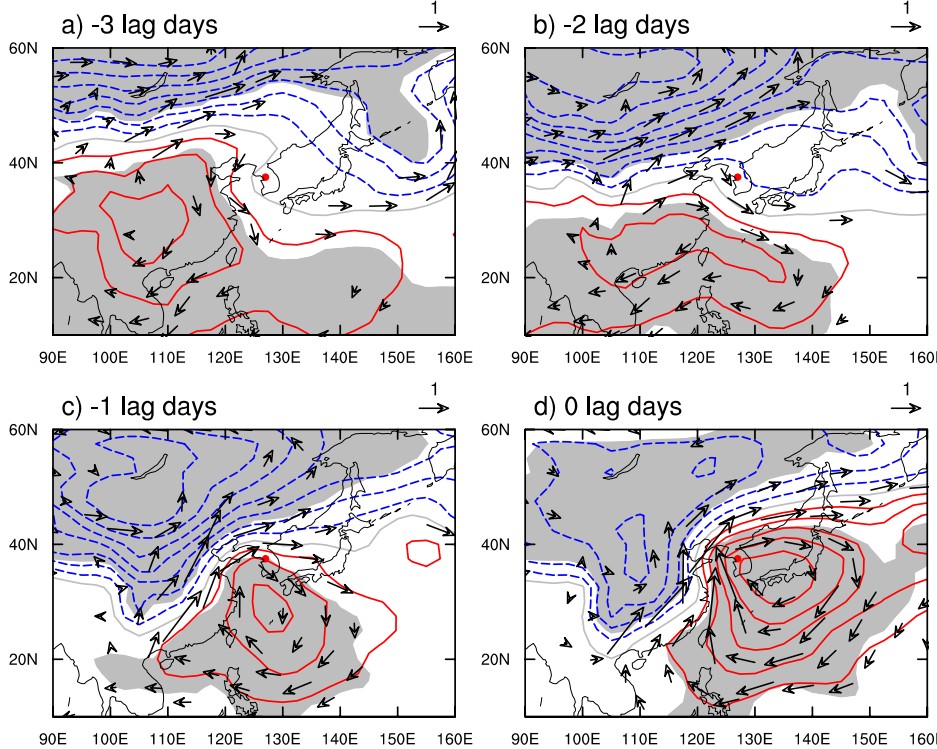

**Figure 3:** Lag composites of seasonal anomalies of geopotential height (contours with interval of 2.5 gpm) and wind (arrows with reference scale of 1 m s$^{-1}$) at 850 hPa relative to each first day that $\exp(\mathrm{PM_{10_{ST}}})$ exceeds the sum of its mean and one standard deviation. Total number of events, mean, and standard deviation are 283, 1, and 0.50, respectively. Geopotential height and wind statistically significant at 95% confidence level are represented as light gray shading and arrows. Location of Seoul is marked by solid red circles.



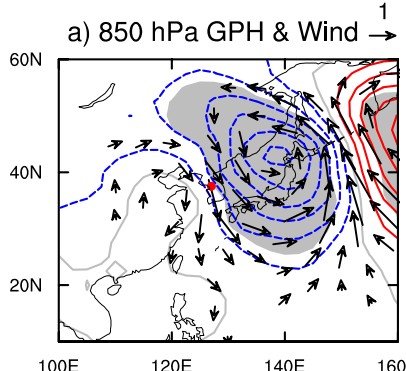

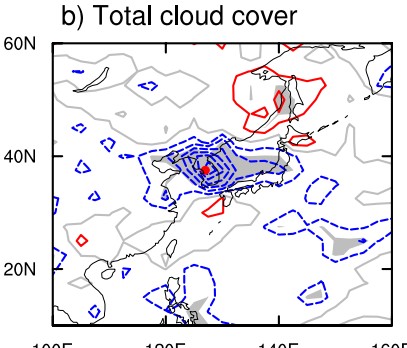

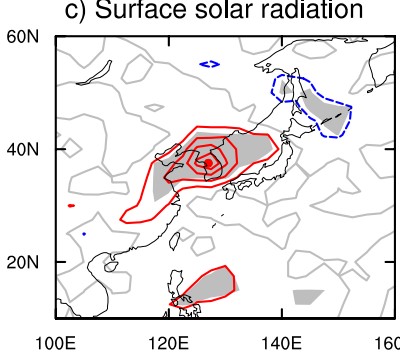

**Figure 4: Composites of seasonal anomalies of (a) geopotential height (contours with interval of 2.5 gpm) and wind (arrows with reference scale of 1 m s$^{-1}$) at 850 hPa, (b) total cloud cover fraction (contours with interval of 2%), (c) downward solar radiation anomaly at surface (contours with interval of 5 W m$^{-2}$) for each first day that $\exp\left(O_{3\,8h_{ST}}\right)$ exceeds sum of the mean and one standard deviation. Total number of events, mean, and standard deviation are 346, 1, and 0.41, respectively. Geopotential height and wind, total cloud cover, and downward solar radiation statistically significant at 95% confidence level are represented by light gray shading and arrows. Location of Seoul is marked by a solid red circle.**



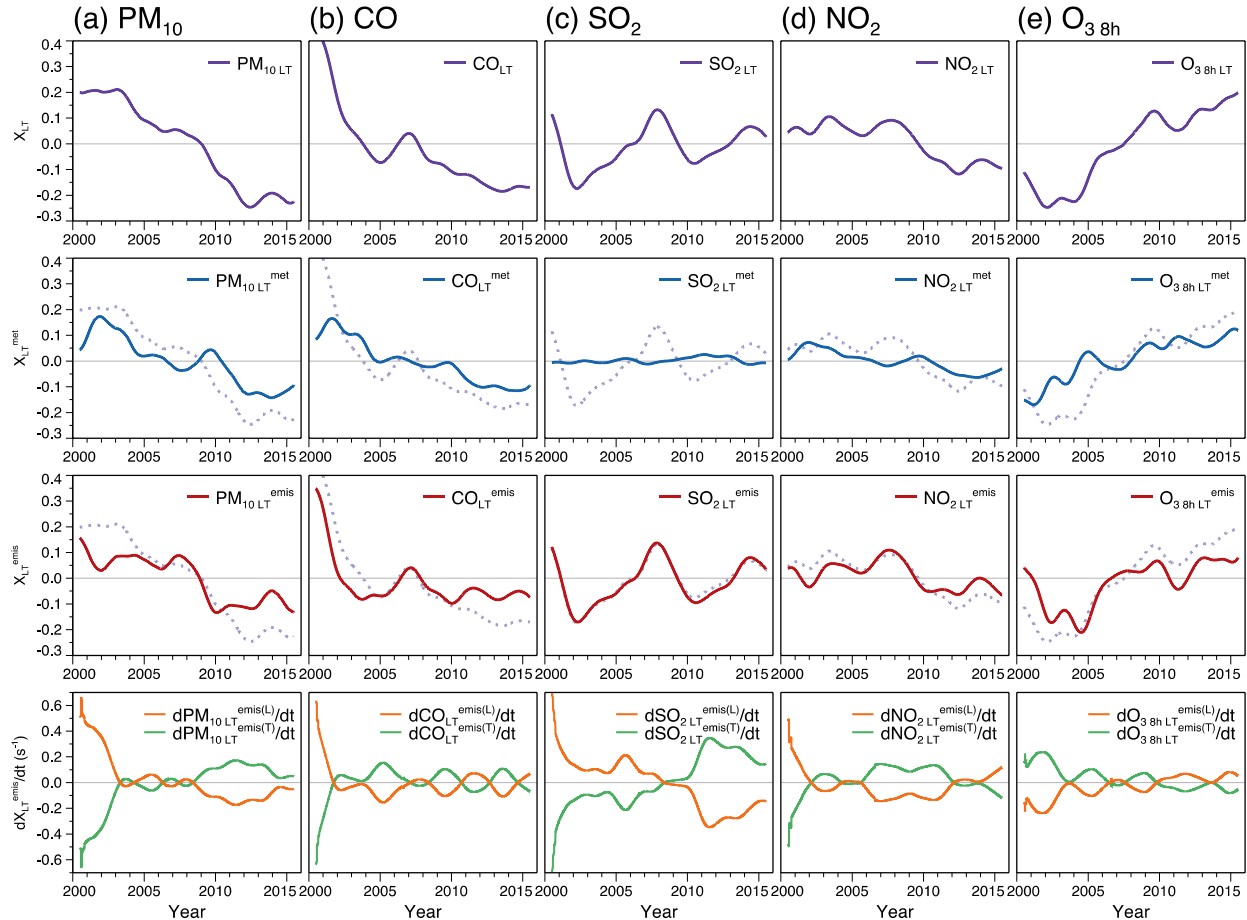

**Figure 5: Long-term components those are unadjusted for the meteorological variables ($X_{LT}$; violet lines), meteorology-related ($X_{LT}^{met}$; blue lines) and emission-related ($X_{LT}^{emis}$; red lines) long-term components, and contributions of local emissions ($\frac{\partial}{\partial t}X_{LT}^{emis(L)}$; orange lines) and transport of regional emissions ($\frac{\partial}{\partial t}X_{LT}^{emis(T)}$; green lines) to the long-term trends of (a) PM$_{10}$, (b) CO, (c) SO$_2$, (d) NO$_2$, and (e) O$_{3\ 8h}$ in Seoul.**



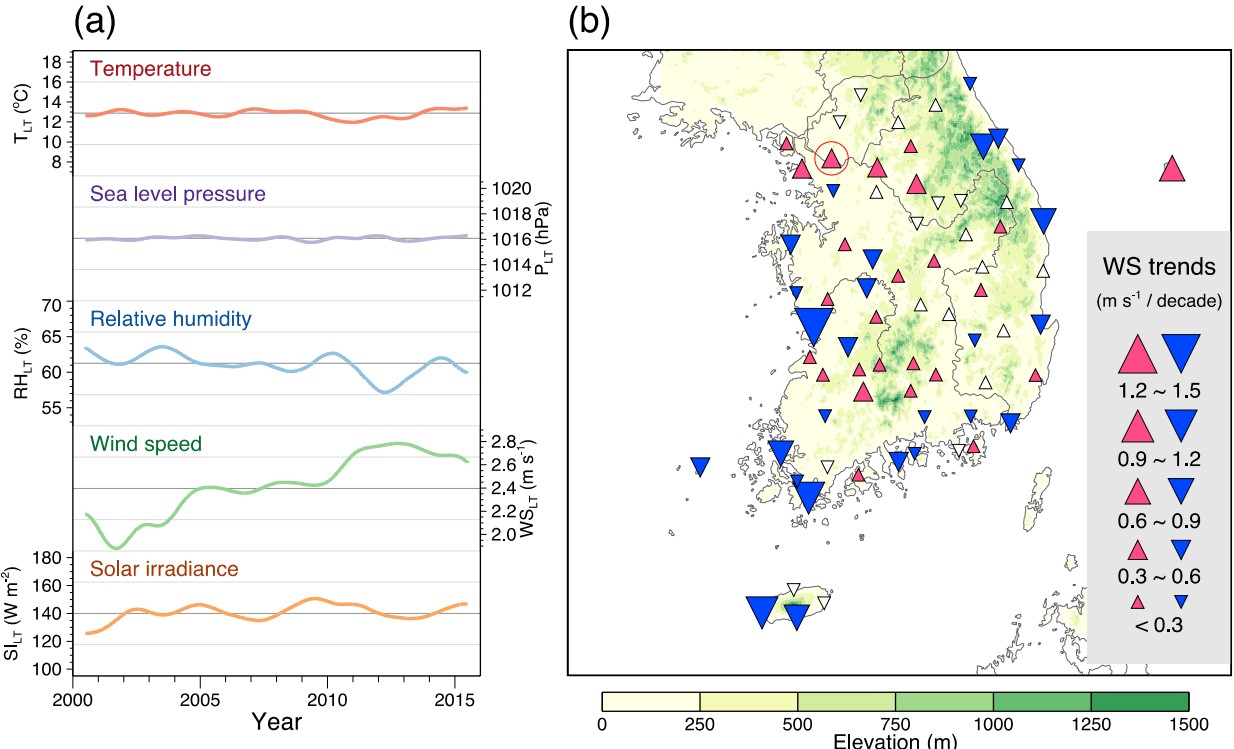

**Figure 6: (a) Long-term components of daily average temperature, sea level pressure, relative humidity, wind speed, and solar irradiance. Vertical scales represented by grey horizontal lines are equivalent to 0.3 standard deviations of each original time series of the meteorological variables. (b) Linear trends of wind speed observed at 75 weather station over South Korea for the period of 1999–2016. Upward and downward triangles with colors indicate the increasing and decreasing trends, which are statistically significant at 95% or more ($p < 0.05$), respectively. Location of Seoul is marked by a red open circle.**



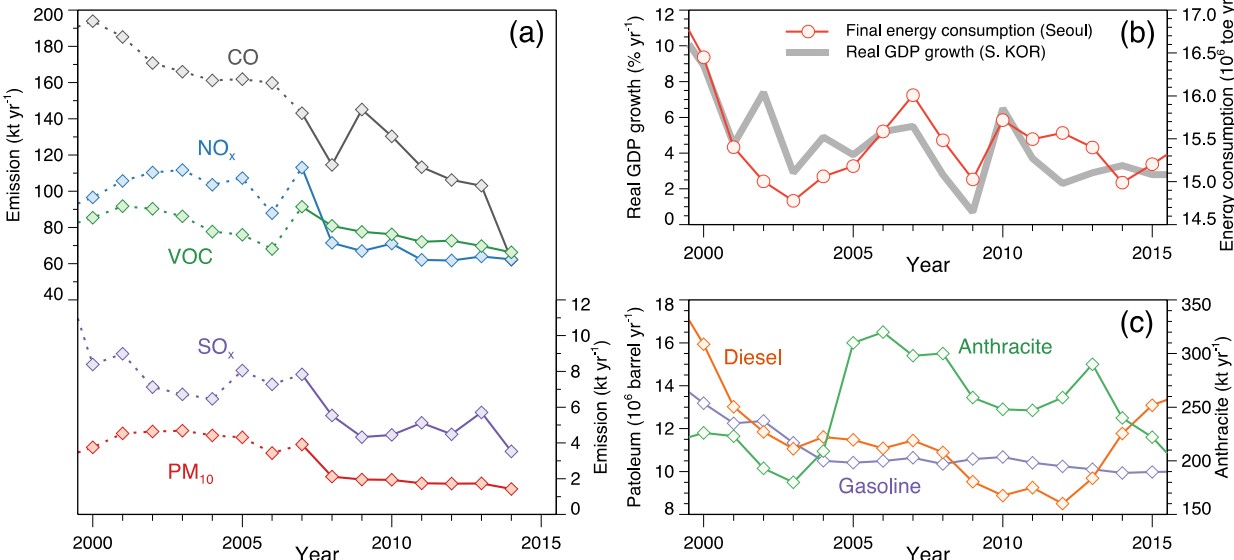

**Figure 7: (a) Annual emissions of SO$_x$, NO$_x$, CO, VOCs, and PM$_{10}$ emissions in Seoul. Note that estimation method for the CAPSS inventory has been continuously updated, but the significant update was made in the year 2007. Emissions before and after 2007 were distinguished by dotted and solid lines. (b) Final energy consumption in Seoul and real GDP growth of South Korea. (c) Diesel, gasoline, and anthracite consumptions in Seoul.**