# Peer review of "Effects of meteorology and emissions on urban air quality: a quantitative statistical approach to long-term records (1999–2016) in Seoul, South Korea"

_Atmospheric Chemistry and Physics, 2018_

## Referee Comment (RC1) · Anonymous Referee #2 · 26 Jul 2018

In the manuscript, the authors used various statistical tools such as the K-Z filter and multiple linear regression method to the 18 year long-term data of the criteria air pollutants and meteorological variables. They could separate short term variations and long-term variation. Further, out of the long-term trend, they could separate the meteorological and emission driven trends. In addition they calculated local emission driven and transported components from the emission driven part based on the continuity equation approach.

It is a well organized manuscript and the results are of great importance since this

study result can be complementary to the 3-dimensional chemical transport modeling results and, thus, it can provide a scientific background for effective policy development in South Korea.

However, there are several points that should be improved and clarified. Thus, I recommend the manuscript be accepted for the publication in the Journal with minor revisions. Specific points are:

1. Abstract needs revision to further emphasize the scientific significances of the study. 2. It would make the manuscript more valuable to compare the results with other the study results for South Korea in which the same statistical tools have been used (for example, Shin et al., AAQR, 12, 93, 2012). Also, it would be nice to refer and compare the study results on the influence of meteorological parameters on the air pollutant level (for example, Lim et al., JKOSAE, 28, 325, 2012). 3. In table 1, add a part on explaining why the sum of the variances is not 100%. Also, in table 4, it would better to make the sum of the trends of 'emis' and 'met' be equal to the total. 4. In eq. (9), should it be a residual term or SLT contains all terms except advection? 5. What would be the main reason for CO(LT,Met) and SO2(LT,Met) to show different trends though both of them are primary air pollutants? 6. The different trends of NO2 and NOx might be caused not only by the reasons explained in the manuscript but also with changing oxidative potential of the atmosphere (for example, Kim and Lee, AAQR, in press).

---

## Referee Comment (RC2) · L. Henneman (Referee) · 15 Aug 2018

**General comments:**

Seo et al. present an evaluation of trends at various frequencies in Seoul, Korea. They use meteorological detrending techniques that apply the KZ filter and multiple linear regressions along with a simplified continuity formulation to attribute long term changes to local and transported sources.

Overall, the analysis is vigorous, and the conclusions appear relevant to future policy decisions. Other than a deeper discussion of the continuity approach and the minor suggestions below, I believe the overall analysis to be sound and an important contribution to the published literature.

**Specific comments:**

The meteorological detrending approach has been applied in similar fashion in previous applications, but I believe the continuity-based derivation of local/transported emissions on the long-term trend to be innovative. My major comments for this manuscript revolve around the development and discussion of this approach—I have listed questions here that I hope will inspire the authors to consider and discuss the approach in more detail. While the interpretation of the results appears to fit within scientific understanding of current atmospheric processes and emissions trends, I think the manuscript would be greatly improved by further development of this method.

1. Please clarify that local/long term transported emissions to long-term trends only are available to Seoul-wide data, not on individual sites. What are the implications of the distance scales, numbers of monitors available, and the choice of centering the cartesian coordinates at the weather station? As the distance scale used here is on the order $10^1$ km, can we assume the origin of the transported pollutants to be a certain distance away (e.g., on the order of $10^2$ km)?
2. What is the interpretation of the high nonlinearities in the meridional gradients (Figure S5)?
3. Are the meridional slopes generally statistically significantly different from zero?
4. Are there limitations to this approach in regard to fewer available monitoring locations, spatial distributions of monitoring sites, etc.? High levels of missing data on certain days could severely impact calculated meridional slopes. Did the authors find evidence of this? If so, was anything done to correct for it?

**Technical comments:**

Page 4, line 9: How were Asian Dust days identified?

Throughout the manuscript (and especially in the Data section), please clarify the monitoring station being referred to, or whether the data is an average of all monitoring stations.

Eqns. 2 & 4: I recommend changing the syntax of these equations slightly to improve clarification. The current form of, e.g., $KZ_{(m,p)}X(t)$ looks like the KZ term is being multiplied by $X(t)$. I recommend changing to $KZ_{(m,p)}[X(t)]$ or similar

Eqns. 10 & 11: Please include section (possibly in the supplement) with more detailed derivation steps.

Page 8, Line 15: I believe there is a typo in this sentence. Possibly it should be "…not balanced *against* each other in *the* short-term timescale"

Page 8, Section 4: Please provide explanation for why the variances described by each of the trends do not sum to 100%.

Page 9, line 22, suggest remove final word ("were")

Page 12, Line 9: Why do you use $p < 0.1$ here, and 0.05 elsewhere?

Page 12, Line 29: I advise referencing the relevant emissions changes in Figure 7 to more fully describe impacts potential impacts of the Legislation

Table 4: Relatively higher (positive or negative) correlation between some met variable ST trends (SI and RH, for example) may affect the interpretation of

Figure S3: Please clarify which monitoring site this data is from, or if it is an average. This clarification should be made throughout the manuscript

Figure S4: I recommend stating that the gray line in each subfigure represents the raw spectrum.

---

## Author Comment (AC1) · 18 Oct 2018

**Response to Anonymous Referee #2**

In the manuscript, the authors used various statistical tools such as the K-Z filter and multiple linear regression method to the 18 year long-term data of the criteria air pollutants and meteorological variables. They could separate short term variations and long-term variation. Further, out of the long-term trend, they could separate the meteorological and emission driven trends. In addition they calculated local emission driven and transported components from the emission driven part based on the continuity equation approach.

It is a well organized manuscript and the results are of great importance since this study result can be complementary to the 3-dimensional chemical transport modeling results and, thus, it can provide a scientific background for effective policy development in South Korea.

However, there are several points that should be improved and clarified. Thus, I recommend the manuscript be accepted for the publication in the Journal with minor revisions. Specific points are:

We appreciate the reviewer's careful reading, valuable comments, and constructive suggestions. We modified the original manuscript as following the reviewer's comments and suggestions. Each response to the reviewer colored in blue and changes in the manuscript colored in red.

1. Abstract needs revision to further emphasize the scientific significances of the study.

We revised the last sentence of the abstract (L27–29 on p.1) as follows:

The present results not only reveal an important role of synoptic meteorological conditions on the episodic air pollution events but also give insights into the practical effects of environmental policies and regulations on the long-term air pollution trends. As a complementary approach to the chemical transport modeling, this study will provide a scientific background for developing and improving effective air quality management strategy in Seoul and its metropolitan area.

2. It would make the manuscript more valuable to compare the results with other the study results for South Korea in which the same statistical tools have been used (for example, Shin et al., AAQR, 12, 93, 2012). Also, it would be nice to refer and compare the study results on the influence of meteorological parameters on the air pollutant level (for example, Lim et al., JKOSAE, 28, 325, 2012).

Thanks for the suggestion. To add a brief comparison with the previous results from Shin et al. (2012), we modified L2–4 on p.14 as follows:

In terms of $O_3$, positive $\frac{\partial}{\partial t}\left(O_{3\ 8h_{LT}}^{emis(T)}\right)$ during the 2000s (Fig. 5e) implies that the transport of regional background $O_3$ played an important role in the meteorologically-adjusted long-term $O_{3\ 8h}$ trend ($O_{3\ 8h_{LT}}^{emis}$) during the period. This result is consistent with the previous study on $O_3$ in Seoul for 2002–2006 using the OZone Isopleth Plotting Package for Research (OZIPR) and the KZ filter (Shin et al., 2012). However, because $O_{3\ 8h_{LT}}^{emis(L)}$ has been gradually changed from the decreasing phase to the increasing phase over the analysis period (Fig. 5e), the recent changes in $O_{3\ 8h_{LT}}^{emis}$ in the 2010s are probably more related to the local secondary production rather than the background $O_3$ transport.

Also, a short discussion of comparison with the dispersion effect of winds on $PM_{10}$ and $NO_2$ levels reported by Lim et al. (2012) is inserted after L11 on p.12 as follows:

The long-term component of wind speed ($WS_{LT}$) in Seoul was increased by ~0.9 m s$^{-1}$ from 2002 to 2012 (Fig. 6a). Previous statistical research on the dispersion effects of winds reported that the fraction of concentrations reduced by wind speed rising of 2 m s$^{-1}$ to 3 m s$^{-1}$ was ~24% for $PM_{10}$ and ~9% for $NO_2$ (Lim et al., 2012). These ratios are comparable to the decrease of meteorology-related long-term concentrations

$(\chi_{\mathrm{LT}}^{\mathrm{met}})$ in PM$_{10}$ and NO$_2$ by ~30% and ~10% for the period, respectively ($\Delta X_{\mathrm{LT}}^{\mathrm{met}}$ ($= \Delta\chi_{\mathrm{LT}}^{\mathrm{met}} / \chi_{\mathrm{LT}}^{\mathrm{met}}$) is −0.3 for PM$_{10}$ and −0.1 for NO$_2$; Fig. 5a and d).

3. In Table 1, add a part on explaining why the sum of the variances is not 100%. Also, in Table 4, it would better to make the sum of the trends of 'emis' and 'met' be equal to the total.

Explained variances in Table 1 are identical to coefficients of determination ($R^2$) between the original time series ($X$) and each decomposed component ($X_{\mathrm{ST}}$, $X_{\mathrm{SN}}$, and $X_{\mathrm{LT}}$), which represent how much of the variability of $X$ is accounted for by the variability of each component. The explained variances of the decomposed time series should sum to 100%, if $X_{\mathrm{ST}}$, $X_{\mathrm{SN}}$, and $X_{\mathrm{LT}}$ were completely independent of each other. However, because the components decomposed by the KZ filter still have some minor correlations among them albeit very weak, the explained variance of each component ($R^2$ with $X$ in percentage) are contributed not only by the variance itself but also by the covariances with other components.
In the revised version, to clarify this, we replaced the explained variances in the original Table 1 with the proportions of each variance and covariance to the total variances. The sum of variances of and covariances among the short-term, seasonal, and long-term components are exactly the same as the total variances of the original time series (see the formula in the revised Table 1).
Following the modified Table 1, "(~50–70%)" in L19 on p.8 is now "(~46–68%)." Also, the original version of Table 1 is modified to the $R^2$ values and is now added to the supplement as Table S2.

**Table 1: Total variances ($Var(X)$) of log-scale times series for Seoul average concentrations of PM$_{10}$, SO$_2$, NO$_2$, CO, and O$_{3\,8h}$, and relative contributions (%) of variances of and covariances ($Cov$) among each component to $Var(X)$. Daily data for the period of July 2000 to Jun 2015 that $X_{\mathrm{LT}}$ data is available were used.**

|  | PM$_{10}$ | SO$_2$ | NO$_2$ | CO | O$_{3\,8h}$ |
|---|---|---|---|---|---|
| $Var(X)$[a] | 0.2998 | 0.1345 | 0.1400 | 0.1540 | 0.4068 |
| $Var(X_{\mathrm{ST}})$ | 63.98% | 48.92% | 67.94% | 46.19% | 42.96% |
| $Var(X_{\mathrm{SN}})$ | 22.07% | 40.58% | 23.86% | 34.66% | 47.06% |
| $Var(X_{\mathrm{LT}})$ | 8.74% | 4.50% | 3.64% | 14.32% | 5.00% |
| $Cov(X_{\mathrm{ST}}, X_{\mathrm{SN}})$ | 2.91% | 2.53% | 2.20% | 2.00% | 2.17% |
| $Cov(X_{\mathrm{SN}}, X_{\mathrm{LT}})$ | −0.29% | 0.47% | 0.09% | 0.42% | 0.32% |
| $Cov(X_{\mathrm{ST}}, X_{\mathrm{LT}})$ | −0.01% | 0.00% | −0.01% | 0.00% | 0.01% |
| $Var(X_{\mathrm{LT}}^{\mathrm{emis}})$ | 2.49% | 4.88% | 1.80% | 4.97% | 1.90% |
| $Var(X_{\mathrm{LT}}^{\mathrm{met}})$ | 2.85% | 0.09% | 1.08% | 4.45% | 1.52% |
| $Cov(X_{\mathrm{LT}}^{\mathrm{emis}}, X_{\mathrm{LT}}^{\mathrm{met}})$ | 1.70% | −0.23% | 0.38% | 2.45% | 0.79% |

[a] Values of variances of each log-scale concentration time series
$Var(X) = Var(X_{\mathrm{ST}}) + Var(X_{\mathrm{ST}}) + Var(X_{\mathrm{ST}}) + 2[Cov(X_{\mathrm{ST}}, X_{\mathrm{SN}}) + Cov(X_{\mathrm{SN}}, X_{\mathrm{LT}}) + Cov(X_{\mathrm{ST}}, X_{\mathrm{LT}})]$

**Table S2. Coefficients of determination ($R^2$) between each component and original time series ($X$) for Seoul average concentrations of PM$_{10}$, SO$_2$, NO$_2$, CO, and O$_{3\,8h}$. Daily data for the period of July 2000 to Jun 2015 that $X_{\mathrm{LT}}$ data is available were used.**

| Components | PM$_{10}$ | SO$_2$ | NO$_2$ | CO | O$_{3\,8h}$ | Notes |
|---|---|---|---|---|---|---|
| $X_{\mathrm{ST}}$ | 0.699 | 0.541 | 0.724 | 0.503 | 0.474 | Short-term components |
| $X_{\mathrm{SN}}$ | 0.276 | 0.468 | 0.287 | 0.397 | 0.522 | Seasonal components |
| $X_{\mathrm{LT}}$ | 0.081 | 0.055 | 0.038 | 0.152 | 0.057 | Long-term components |
| $X_{\mathrm{LT}}^{\mathrm{emis}}$ | 0.064 | 0.054 | 0.030 | 0.124 | 0.043 | Emission-related $X_{\mathrm{LT}}$ |
| $X_{\mathrm{LT}}^{\mathrm{met}}$ | 0.069 | 0.003 | 0.019 | 0.107 | 0.040 | Meteorology-related $X_{\mathrm{LT}}$ |

In Table 4, the sum of the linear trends of $X_{LT}^{emis}$ and $X_{LT}^{met}$ should be exactly the same as the $X_{LT}$ trend. However, the rounding after the calculation that we used here can induce tiny difference of ±1 in the last decimal place of both $\frac{\partial X_{LT}}{\partial t}$ and the sum of $\frac{\partial}{\partial t}X_{LT}^{emis}$ and $\frac{\partial}{\partial t}X_{LT}^{met}$. For example, $PM_{10_{LT}}^{emis}$ trend ($-1.6883300\%$ yr$^{-1}$) and $PM_{10_{LT}}^{met}$ trend ($-1.9456153\%$ yr$^{-1}$) exactly sum to $PM_{10_{LT}}$ trend ($-3.6339453\%$ yr$^{-1}$). However, the trends rounded to the second decimal place are $-1.69\%$ yr$^{-1}$ for $PM_{10_{LT}}^{emis}$ and $-1.95\%$ yr$^{-1}$ for $PM_{10_{LT}}^{met}$, and thus their sum is $0.01\%$ yr$^{-1}$ larger than the rounded-off $PM_{10_{LT}}$ trend ($-3.63\%$ yr$^{-1}$). For the result value, we think that the rounding after the calculation is more accurate than the rounding before the calculation. Therefore, we left Table 4 as the original version.

4. In eq. (9), should it be a residual term or $S_{LT}$ contains all terms except advection?

By Eq. (9), the term $S_{LT}$ should contain all effects on the change rate of long-term trend ($\frac{\partial X_{LT}}{\partial t}$), except the changes of inflow and outflow by horizontal advection ($-\vec{V}_{LT} \cdot \nabla X_{LT}$). The residual effects are source (surface emissions and atmospheric secondary production related to $X_{LT}^{emis(L)}$ and partly $X_{LT}^{met}$), sink (physical and chemical scavenging affected by $X_{LT}^{met}$), and diffusion (accumulation and dissipation), as well as the vertical advection term ($-w_{LT}\frac{\partial X_{LT}}{\partial z}$). The term $-w_{LT}\frac{\partial X_{LT}}{\partial z}$ is negligible because the long-term trend of vertical motion ($w_{LT}$) is close to zero. Therefore, $S_{LT}$ can be regarded as the sum of source term (directly related to $\frac{\partial}{\partial t}X_{LT}^{emis(L)}$) and sink and diffusion terms (affected by the long-term meteorological effect; $\frac{\partial}{\partial t}X_{LT}^{met}$).

5. What would be the main reason for CO(LT,Met) and SO$_2$(LT,Met) to show different trends though both of them are primary air pollutants?

The local meteorological factors such as wind speed, relative humidity, insolation, and temperature are not directly related to the regional transport of air pollutants, and thus can affect mainly on the contribution of local source pollutants rather than transported pollutants. In other words, if the local emissions at a local place are assumed to be zero, the local meteorological effects on accumulation and secondary production ($X_{LT}^{met}$) in the long-term air pollution trend ($X_{LT}$) will be negligible. As we wrote in the third paragraph on p.11, the emission intensity of CO in Seoul (215 t/km$^2$) is higher than those in the upwind source areas (e.g., 98 t/km$^2$ for the Chinese eastern coast), while that of SO$_2$ in Seoul (7 t/km$^2$) is much lower compared to those in the upwind source areas (e.g., 14 t/km$^2$ for the Chinese eastern coast). Therefore, the long-term SO$_2$ trend ($SO_{2_{LT}}$) is much less contributed by the changes in meteorological effects on its local emission ($SO_{2_{LT}}^{met}$) than by the changes in its regional emissions ($SO_{2_{LT}}^{emis}$). As we describe in L9–13 on p.13, the interannual variation of satellite-based SO$_2$ level over China is similar to $SO_{2_{LT}}$ (and $SO_{2_{LT}}^{emis}$) in this study (Fig. 5c) and supports the large influence of the regional background SO$_2$ on the long-term SO$_2$ trend in Seoul.

6. The different trends of NO$_2$ and NO$_x$ might be caused not only by the reasons explained in the manuscript but also with changing oxidative potential of the atmosphere (for example, Kim and Lee, AAQR, in press).

We appreciate the reviewer's comment. L8–L12 on p.11 was now modified as follows:

Interestingly, NO$_2$ level has been stabilized despite increasing of the number of vehicles in Seoul from 2.3 million in 1999 to 3.1 million in 2016 probably owing to implementation of natural gas vehicles and low emission diesel engines, and NO$_x$ (= NO + NO$_2$) level has been even decreased from ~70 ppb to ~50 ppb for the same period (Shon and Kim, 2011; Kim and Lee, 2018). Such an increase of NO$_2$ to NO$_x$ ratio implies that additional conversion of NO to NO$_2$ occurs somewhere before the emission (e.g., exhaust line of the vehicle) or in the atmosphere. Although further evidence is required, this can be attributable to expanding of diesel particulate filter (DPF) and diesel oxidation catalyst (DOC) usage for diesel vehicles or increase of the atmospheric oxidative potential in the SMA (Alvarez et al, 2008; Kim and Lee, 2018).

---

## Author Response (AR1)

**Response to Referee #1 (Dr. Lucas Henneman)**

**General comments:**
Seo et al. present an evaluation of trends at various frequencies in Seoul, Korea. They use meteorological detrending techniques that apply the KZ filter and multiple linear regressions along with a simplified continuity formulation to attribute long term changes to local and transported sources.
Overall, the analysis is vigorous, and the conclusions appear relevant to future policy decisions. Other than a deeper discussion of the continuity approach and the minor suggestions below, I believe the overall analysis to be sound and an important contribution to the published literature.

We appreciate the reviewer for careful reading and helpful comments that improve the quality of the manuscript. As indicated in the following point-by-point responses, we have incorporated the reviewer's comments and suggestions into the revised manuscript. Each response to reviewer colored in blue and changes in the manuscript colored in red.

**Specific comments:**
The meteorological detrending approach has been applied in similar fashion in previous applications, but I believe the continuity-based derivation of local/transported emissions on the long-term trend to be innovative. My major comments for this manuscript revolve around the development and discussion of this approach—I have listed questions here that I hope will inspire the authors to consider and discuss the approach in more detail. While the interpretation of the results appears to fit within scientific understanding of current atmospheric processes and emissions trends, I think the manuscript would be greatly improved by further development of this method.

1. Please clarify that local/long term transported emissions to long-term trends only are available to Seoul-wide data, not on individual sites.

As the reviewer pointed out, the evaluation method for the changes in local emissions ($X_{LT}^{emis(L)}$) and the changes in transport of regional emissions ($X_{LT}^{emis(T)}$) we suggested here is unavailable for individual air quality monitoring sites because the horizontal advection was obtained by using the winds at a weather station and the horizontal gradient of air pollutants measured in the surrounding area. To clarify this, we add the following sentence at the end of L24 on p.7.

Note that the above method to evaluate the changes in $X_{LT}^{emis(T)}$ and $X_{LT}^{emis(L)}$ is applicable not to data from an individual site but to data from the wide area, because of the requirement of horizontal gradient term in Eq. (10).

What are the implications of the distance scales, numbers of monitors available, and the choice of centering the Cartesian coordinates at the weather station?

As we described in Sect. 2, we selected 18 air quality monitoring sites within the Seoul area and took average the daily data from the selected sites to produce the daily city-average data. Since the selected 18 monitoring sites in Seoul are located within the area of ~15 km radius from the weather station (37.571°N, 126.966°E; Fig. 1a), we can assume a 30 km grid box centered at the weather station, which is representative for the daily air pollution data in Seoul. If we try to calculate the horizontal advection of the long-term components ($-\vec{V}_{LT} \cdot \nabla X_{LT}$) at the center of the coordinate (the $(i, j)$th grid point), the long-term component of $u$- and $v$-wind data at the $(i, j)$th grid point and $X_{LT}$ at the $(i-1, j)$th, $(i+1, j)$th, $(i, j-1)$th, and $(i, j+1)$th grid points will be required as follows:

$$\left(-\vec{V}_{LT} \cdot \nabla X_{LT}\right)_{(i,j)} = -u_{LT(i,j)} \frac{X_{LT(i+1,j)} - X_{LT(i-1,j)}}{2\Delta x} - v_{LT(i,j)} \frac{X_{LT(i,j+1)} - X_{LT(i,j-1)}}{2\Delta y}$$

where $\Delta x$ and $\Delta y$ are 30 km, and thus the total area we need for the calculation will be 90 km × 90 km (3 grid boxes each for each $x$- and $y$-direction). Although $\nabla X_{LT}$ was obtained by using the linear regression method

in this study because of scattered monitoring sites over the Seoul Metropolitan Area (SMA), we partly adopted the finite difference concept to set up the center of coordinates (to the Seoul weather station) and specified the area for the available sites (as a 50 km radius range from the center of coordinates).

The number of monitoring sites used here was decided by setting a criterion for data availability at each site, which is a ratio of the available number of data ($N_{AD}$) to the total number of data ($N_{TD}$). As shown in Fig. S8a, the number of available sites ($N_{AS}$) is rapidly decreased where the data availability is larger than 75% ($N_{AD}$ / $N_{TD} > 0.75$). Because both $N_{AS}$ and $N_{AD}$ / $N_{TD}$ are important, we examined the multiplication of $N_{AS}$ and $N_{AD}$ / $N_{TD}$ as a score and found that the score was high enough when the criterion of data availability was 75% (Fig. S8b). Fig. S8 was now added to the supplement.

[Figure]

Figure S8. (a) Numbers of available air quality monitoring sites ($N_{AS}$) within the area of 50 km radius from the Seoul weather station and (b) $N_{AS}$ multiplied with ratios of the available number of data ($N_{AD}$) to the total number of data ($N_{TD}$). Vertical dotted lines at $N_{AD}$ / $N_{TD}$ of 0.75 show the data availability of 75% and a horizontal dotted line at $N_{AS}$ of 70 represents the number of air quality monitoring sites used for obtaining the horizontal gradient of long-term components in this study.

As the distance scale used here is on the order $10^1$ km, can we assume the origin of the transported pollutants to be a certain distance away (e.g., on the order of $10^2$ km)?

In this study, we investigated the changes in transport of regional emissions ($\frac{\partial}{\partial t} X_{LT}^{emis(T)}$) using the data within the area of the radius of 50 km from the Seoul weather station. Therefore, the interpretation of the result should be limited within the analysis area. Considering that the interpretable area almost agrees with the Seoul Metropolitan Area (SMA), the origins of the transported pollutants from the outside of Seoul can not only be the outside of the SMA (such as the Chinese eastern coasts) but also be the inside of the SMA (such as industrial areas in the southwestern SMA).

2. What is the interpretation of the high nonlinearities in the meridional gradients (Figure S5)?

The nonlinearity of the slope in the scatter plot of $X_{BL}$ versus $y$-axis ($r = -0.385$ and $p < 0.001$; Fig. S5b) mainly arises from the overall zonal gradient of $X_{BL}$ ($\frac{\partial X_{BL}}{\partial x} = -0.0017$ km$^{-1}$; Fig. S5c). Similarly, the nonlinearity of the slope in the scatter plot of $X_{BL}$ versus $x$-axis ($r = -0.202$ and $p = 0.094$; Fig. S5c) mostly

results from the meridional gradient of $X_{BL}$ over the analysis area ($\frac{\partial X_{BL}}{\partial y} = -0.0019$ km$^{-1}$; Fig. S5b). Since $\left|\frac{\partial X_{BL}}{\partial y}\right| > \left|\frac{\partial X_{BL}}{\partial x}\right|$, the nonlinearity of the slope in Fig. S5c ($X_{BL}$ vs. $x$) is larger than that in Fig. S5b ($X_{BL}$ vs. $y$).

3. Are the meridional slopes generally statistically significantly different from zero?

The zonal and meridional slopes for the long-term components are statistically significant when the slope is high enough. We modified Fig. S6 and Fig. 5 to reveal the statistically significant ($p < 0.05$) slope of each long-term component ($X_{LT}$) and changes in local emissions ($\frac{\partial}{\partial t}\left(X_{LT}^{emis(L)}\right)$) and transport ($\frac{\partial}{\partial t}\left(X_{LT}^{emis(T)}\right)$).

[Figure]

**Figure S6.** Long-term component of (a) zonal wind ($u_{LT}$) and (b) meridional wind ($v_{LT}$) at the Seoul weather station. Zonal gradients ($\partial X_{LT}/\partial x$, red lines) and meridional gradients ($\partial X_{LT}/\partial y$, blue lines) of the long-term components and transport terms ($-\vec{V}_{LT} \cdot \nabla X_{LT}$, violet lines) by long-term components of horizontal winds ($\vec{V}_{LT} = (u_{LT}, v_{LT})$) for (c–d) PM$_{10}$, (e–f) CO, (g–h) SO$_2$, (i–j) NO$_2$, (k–l) O$_{3\,8h}$. Solid lines in horizontal gradients and transport terms indicate that the gradients obtained by linear regression are statistically significant at the 95% level or higher ($p < 0.05$).

[Figure]

**Figure 5:** Long-term components those are unadjusted for the meteorological variables ($X_{LT}$; violet lines), meteorology-related ($X_{LT}^{met}$; blue lines) and emission-related ($X_{LT}^{emis}$; red lines) long-term components, and contributions of local emissions ($\frac{\partial}{\partial t}X_{LT}^{emis(L)}$; orange lines) and transport of regional emissions ($\frac{\partial}{\partial t}X_{LT}^{emis(T)}$; green lines) to the long-term trends of (a) $PM_{10}$, (b) CO, (c) $SO_2$, (d) $NO_2$, and (e) $O_{3\,8h}$ in Seoul. Solid lines in $\frac{\partial}{\partial t}X_{LT}^{emis(L)}$ and $\frac{\partial}{\partial t}X_{LT}^{emis(T)}$ show that the horizontal gradients of $X_{LT}$ ($-\vec{V}_{LT} \cdot \nabla X_{LT}$) obtained by linear regression are statistically significant at the 95% level or higher ($p < 0.05$).

Although the interpretable periods for each rate of changes in $X_{LT}^{emis(L)}$ and $X_{LT}^{emis(T)}$ are reduced based on the statistical significance test ($p < 0.05$), important features we described in Sect. 4.3.2 still remain in the modified version of Fig. 5.

4. Are there limitations to this approach in regard to fewer available monitoring locations, spatial distributions of monitoring sites, etc.?

The most severe limitation of this approach arises from the high dependence on both number and spatial distribution of monitoring sites. If the number of sites is not enough for statistical analysis or the distribution of sites is biased from the center of the analysis area, the advection (transport) term of the tracer continuity equation must be unreliable. In addition, there must be wind data at the center of the analysis area.

High levels of missing data on certain days could severely impact calculated meridional slopes. Did the authors find evidence of this? If so, was anything done to correct for it?

In this study, we applied this approach to the long-term components of each air pollutant. Because of the iterative moving average process of the KZ-filter, together with the high data availability criterion for each site (75%), the number of missing data for the long-term components at each site was negligible. However, it

can be easily speculated that the data missing at several sites must affect the slope of data and cause abrupt changes in the advection term.

**Technical comments:**

Page 4, line 9: How were Asian Dust days identified?

We used daily records of the Asian Dust events, which were provided by the Korea Meteorological Administration (KMA) based on both naked eye observations (following the WMO recommendation) and $PM_{10}$ measurements (using the beta attenuation monitoring method). The KMA website is now cited in L9 on p.4 and included in the reference list as follows:

KMA (Korea Meteorological Administration): Asian Dust observation days, available at: http://www.weather.go.kr/weather/asiandust/observday.jsp?type=2&stnId=108&year=2016&x=20&y =11, (last access: 18 October 2018), 2018 (in Korean).

Throughout the manuscript (and especially in the Data section), please clarify the monitoring station being referred to, or whether the data is an average of all monitoring stations.

Because we aimed to analyzed city-scale air pollution variability in Seoul, we averaged daily data for selected 18 sites within the city and used in this study. To clarify, we added the following sentence to the first paragraph of Sect. 2 (L5 on p.4).

We averaged daily concentration data from the selected 18 sites in Seoul and utilized in this study.

Eqns. 2 & 4: I recommend changing the syntax of these equations slightly to improve clarification. The current form of, e.g., $KZ_{(m,p)}X(t)$ looks like the KZ term is being multiplied by $X(t)$. I recommend changing to $KZ_{(m,p)}[X(t)]$ or similar.

We modified Eqs. (2), (4), and (7), and L2 on p.6, following the reviewer's suggestion.

Eqns. 10 & 11: Please include section (possibly in the supplement) with more detailed derivation steps.

We added detailed derivation steps for Eqs. (10) and (11) as Appendix S1 in the supplement.

Page 8, Line 15: I believe there is a typo in this sentence. Possibly it should be "...not balanced *against* each other in *the* short-term timescale"

Thanks for the correction. It was now corrected.

Page 8, Section 4: Please provide explanation for why the variances described by each of the trends do not sum to 100%.

Explained variances in Table 1 are identical to coefficients of determination ($R^2$) between the original time series ($X$) and each decomposed component ($X_{ST}$, $X_{SN}$, and $X_{LT}$), which represent how much of the variability of $X$ is accounted for by the variability of each component. The explained variances of the decomposed time series should sum to 100%, if $X_{ST}$, $X_{SN}$, and $X_{LT}$ were completely independent of each other. However, because the decomposed components by the KZ filter method still have minor correlations among them albeit very weak, the explained variance of each component ($R^2$ with $X$ in percentage) are contributed not only by the variance itself but also by the covariances with other components.
In the revised version, to clarify this, we replaced the explained variances in the original version of Table 1 with the proportions of each variance and covariance to the total variances. The sum of variances of and covariances among the short-term, seasonal, and long-term components are exactly the same as the total variances of the original time series (see the formula in the revised Table 1).

Following the modified Table 1, "(~50–70%)" in L19 on p.8 is now "(~46–68%)." Also, the original version of Table 1 is modified to the $R^2$ values and is now added to the supplement as Table S2.

**Table 1: Total variances ($Var(X)$) of log-scale times series for Seoul average concentrations of $PM_{10}$, $SO_2$, $NO_2$, CO, and $O_{3\ 8h}$, and relative contributions (%) of variances of and covariances ($Cov$) among each component to $Var(X)$. Daily data for the period of July 2000 to Jun 2015 that $X_{LT}$ data is available were used.**

| | $PM_{10}$ | $SO_2$ | $NO_2$ | CO | $O_{3\ 8h}$ |
|---|---|---|---|---|---|
| $Var(X)^a$ | 0.2998 | 0.1345 | 0.1400 | 0.1540 | 0.4068 |
| $Var(X_{ST})$ | 63.98% | 48.92% | 67.94% | 46.19% | 42.96% |
| $Var(X_{SN})$ | 22.07% | 40.58% | 23.86% | 34.66% | 47.06% |
| $Var(X_{LT})$ | 8.74% | 4.50% | 3.64% | 14.32% | 5.00% |
| $Cov(X_{ST}, X_{SN})$ | 2.91% | 2.53% | 2.20% | 2.00% | 2.17% |
| $Cov(X_{SN}, X_{LT})$ | −0.29% | 0.47% | 0.09% | 0.42% | 0.32% |
| $Cov(X_{ST}, X_{LT})$ | −0.01% | 0.00% | −0.01% | 0.00% | 0.01% |
| $Var(X_{LT}^{emis})$ | 2.49% | 4.88% | 1.80% | 4.97% | 1.90% |
| $Var(X_{LT}^{met})$ | 2.85% | 0.09% | 1.08% | 4.45% | 1.52% |
| $Cov(X_{LT}^{emis}, X_{LT}^{met})$ | 1.70% | −0.23% | 0.38% | 2.45% | 0.79% |

$^a$ Values of variances of each log-scale concentration time series

$$Var(X) = Var(X_{ST}) + Var(X_{ST}) + Var(X_{ST}) + 2[Cov(X_{ST}, X_{SN}) + Cov(X_{SN}, X_{LT}) + Cov(X_{ST}, X_{LT})]$$

**Table S2. Coefficients of determination ($R^2$) between each component and original time series ($X$) for Seoul average concentrations of $PM_{10}$, $SO_2$, $NO_2$, CO, and $O_{3\ 8h}$. Daily data for the period of July 2000 to Jun 2015 that $X_{LT}$ data is available were used.**

| Components | $PM_{10}$ | $SO_2$ | $NO_2$ | CO | $O_{3\ 8h}$ | Notes |
|---|---|---|---|---|---|---|
| $X_{ST}$ | 0.699 | 0.541 | 0.724 | 0.503 | 0.474 | Short-term components |
| $X_{SN}$ | 0.276 | 0.468 | 0.287 | 0.397 | 0.522 | Seasonal components |
| $X_{LT}$ | 0.081 | 0.055 | 0.038 | 0.152 | 0.057 | Long-term components |
| $X_{LT}^{emis}$ | 0.064 | 0.054 | 0.030 | 0.124 | 0.043 | Emission-related $X_{LT}$ |
| $X_{LT}^{met}$ | 0.069 | 0.003 | 0.019 | 0.107 | 0.040 | Meteorology-related $X_{LT}$ |

Page 9, line 22, suggest remove final word ("were").

Thanks. It was now corrected.

Page 12, Line 9: Why do you use $p < 0.1$ here, and 0.05 elsewhere?

Here "a significant ($p < 0.1$) linear trend" was intended not to emphasize that "$p < 0.1$" is the significant threshold but to describe that the linear trend of $WS_{LT}$ is statistically significant at 90% confidence level. Although the $p$-value of $WS_{LT}$ is slightly larger than 0.05 ($p = 0.065$), this is much lower than those of other long-term meteorological components (Table 4). In general, the long-term trends have large autocorrelations for long lag-periods and thus very small numbers of independent data points. For example, although the correlation coefficient of the slop of $WS_{LT}$ is very high (0.935), its degree of freedom is only 2 (Table S1) probably related to the two data points of the lowest $WS_{LT}$ in 2002 and the highest $WS_{LT}$ in 2013, and therefore the $p$-value of the slope is not much small ($p = 0.065$).

Page 12, Line 29: I advise referencing the relevant emissions changes in Figure 7 to more fully describe impacts potential impacts of the Legislation.

We appreciate the reviewer's comment. To describe the "Special Act on the Improvement of Air Quality in Seoul Metropolitan Area" and its potential impact on the emission trends in Seoul, following sentences were now added to the end of L2 on p.13.

Although the act includes the introduction of the emission cap-and-trade system and strengthening VOCs management, most of the budget has been allocated for expanding of DPF/DOC usage for old diesel vehicles, which was considered to be effective for reduction of $NO_x$ and primary PM emissions (Kim and Lee, 2018). The impact of such policy on the decrease in $PM_{10}$ and $NO_x$ emissions (Fig. 7a) is not easily distinguishable from the influence of the decrease in diesel consumption (Fig. 7c). However, the generalized use of DPF/DOC in diesel vehicles may be one reason for relatively stable trends of $PM_{10}$ and $NO_x$ emissions despite the rapid increase in diesel consumption mainly by vehicles since 2012.

Table 4: Relatively higher (positive or negative) correlation between some met variable ST trends (SI and RH, for example) may affect the interpretation of

We agree with the reviewer's points. In fact, the correlations among the short-term meteorological components are closely related to the midlatitude synoptic weather system. The positive correlation between P and SI and the negative correlations of RH with P and SI implies clear-sky and dry conditions in the high-pressure system and cloudy and wet conditions in the low-pressure system. In Sect. 4.2, we already considered the interpretation of the correlations among the short-term meteorological components to depict the impact of the synoptic meteorological conditions on the short-term variability of pollutants.

Figure S3: Please clarify which monitoring site this data is from, or if it is an average. This clarification should be made throughout the manuscript.

In this study, we used daily pollutant concentrations averaged for the selected 18 sites in Seoul. We modified captions of Fig. 2, Fig. S1, and Fig. S3 as follows:

Figure 2: Schematic flowchart of temporal decomposition of air pollution time series (Seoul average daily $PM_{10}$ concentration) into short-term, seasonal, and emission-related and meteorology-related long-term components.

Figure S1. (a) Numbers of available air quality monitoring sites in Seoul, of which missing data are less than 10% of the total. (b) Average and (c) standard deviation of $PM_{10}$ concentrations of the available 18 sites in Seoul. Asian dust events those were excluded from the $PM_{10}$ analysis are marked with orange color.

Figure S3. Decompositions of log-scale daily time series of (a) $PM_{10}$ and (b) $O_{3\ 8h}$ those averaged for 18 sites in Seoul.

Figure S4: I recommend stating that the gray line in each subfigure represents the raw spectrum.

We added following sentence at the end of the caption of Fig. S4.

The power spectrum of the original time series in (a) is represented with gray lines in (b-d).

**Response to Anonymous Referee #2**

In the manuscript, the authors used various statistical tools such as the K-Z filter and multiple linear regression method to the 18 year long-term data of the criteria air pollutants and meteorological variables. They could separate short term variations and long-term variation. Further, out of the long-term trend, they could separate the meteorological and emission driven trends. In addition they calculated local emission driven and transported components from the emission driven part based on the continuity equation approach.

It is a well organized manuscript and the results are of great importance since this study result can be complementary to the 3-dimensional chemical transport modeling results and, thus, it can provide a scientific background for effective policy development in South Korea.

However, there are several points that should be improved and clarified. Thus, I recommend the manuscript be accepted for the publication in the Journal with minor revisions. Specific points are:

We appreciate the reviewer's careful reading, valuable comments, and constructive suggestions. We modified the original manuscript as following the reviewer's comments and suggestions. Each response to the reviewer colored in blue and changes in the manuscript colored in red.

1. Abstract needs revision to further emphasize the scientific significances of the study.

We revised the last sentence of the abstract (L27–29 on p.1) as follows:

The present results not only reveal an important role of synoptic meteorological conditions on the episodic air pollution events but also give insights into the practical effects of environmental policies and regulations on the long-term air pollution trends. As a complementary approach to the chemical transport modeling, this study will provide a scientific background for developing and improving effective air quality management strategy in Seoul and its metropolitan area.

2. It would make the manuscript more valuable to compare the results with other the study results for South Korea in which the same statistical tools have been used (for example, Shin et al., AAQR, 12, 93, 2012). Also, it would be nice to refer and compare the study results on the influence of meteorological parameters on the air pollutant level (for example, Lim et al., JKOSAE, 28, 325, 2012).

Thanks for the suggestion. To add a brief comparison with the previous results from Shin et al. (2012), we modified L2–4 on p.14 as follows:

In terms of $O_3$, positive $\frac{\partial}{\partial t}\left(O_{3\ 8h_{LT}}^{emis(T)}\right)$ during the 2000s (Fig. 5e) implies that the transport of regional background $O_3$ played an important role in the meteorologically-adjusted long-term $O_{3\ 8h}$ trend ($O_{3\ 8h_{LT}}^{emis}$) during the period. This result is consistent with the previous study on $O_3$ in Seoul for 2002–2006 using the OZone Isopleth Plotting Package for Research (OZIPR) and the KZ filter (Shin et al., 2012). However, because $O_{3\ 8h_{LT}}^{emis(L)}$ has been gradually changed from the decreasing phase to the increasing phase over the analysis period (Fig. 5e), the recent changes in $O_{3\ 8h_{LT}}^{emis}$ in the 2010s are probably more related to the local secondary production rather than the background $O_3$ transport.

Also, a short discussion of comparison with the dispersion effect of winds on $PM_{10}$ and $NO_2$ levels reported by Lim et al. (2012) is inserted after L11 on p.12 as follows:

The long-term component of wind speed ($WS_{LT}$) in Seoul was increased by ~0.9 m s$^{-1}$ from 2002 to 2012 (Fig. 6a). Previous statistical research on the dispersion effects of winds reported that the fraction of concentrations reduced by wind speed rising of 2 m s$^{-1}$ to 3 m s$^{-1}$ was ~24% for $PM_{10}$ and ~9% for $NO_2$ (Lim et al., 2012). These ratios are comparable to the decrease of meteorology-related long-term concentrations

$(\chi_{LT}^{met})$ in PM$_{10}$ and NO$_2$ by ~30% and ~10% for the period, respectively ($\Delta X_{LT}^{met}$ ($= \Delta\chi_{LT}^{met}/\chi_{LT}^{met}$) is −0.3 for PM$_{10}$ and −0.1 for NO$_2$; Fig. 5a and d).

3. In Table 1, add a part on explaining why the sum of the variances is not 100%. Also, in Table 4, it would better to make the sum of the trends of 'emis' and 'met' be equal to the total.

Explained variances in Table 1 are identical to coefficients of determination ($R^2$) between the original time series ($X$) and each decomposed component ($X_{ST}$, $X_{SN}$, and $X_{LT}$), which represent how much of the variability of $X$ is accounted for by the variability of each component. The explained variances of the decomposed time series should sum to 100%, if $X_{ST}$, $X_{SN}$, and $X_{LT}$ were completely independent of each other. However, because the components decomposed by the KZ filter still have some minor correlations among them albeit very weak, the explained variance of each component ($R^2$ with $X$ in percentage) are contributed not only by the variance itself but also by the covariances with other components.
In the revised version, to clarify this, we replaced the explained variances in the original Table 1 with the proportions of each variance and covariance to the total variances. The sum of variances of and covariances among the short-term, seasonal, and long-term components are exactly the same as the total variances of the original time series (see the formula in the revised Table 1).
Following the modified Table 1, "(~50–70%)" in L19 on p.8 is now "(~46–68%)." Also, the original version of Table 1 is modified to the $R^2$ values and is now added to the supplement as Table S2.

**Table 1: Total variances ($Var(X)$) of log-scale times series for Seoul average concentrations of PM$_{10}$, SO$_2$, NO$_2$, CO, and O$_{3\ 8h}$, and relative contributions (%) of variances of and covariances ($Cov$) among each component to $Var(X)$. Daily data for the period of July 2000 to Jun 2015 that $X_{LT}$ data is available were used.**

|  | PM$_{10}$ | SO$_2$ | NO$_2$ | CO | O$_{3\ 8h}$ |
|---|---|---|---|---|---|
| $Var(X)^a$ | 0.2998 | 0.1345 | 0.1400 | 0.1540 | 0.4068 |
| $Var(X_{ST})$ | 63.98% | 48.92% | 67.94% | 46.19% | 42.96% |
| $Var(X_{SN})$ | 22.07% | 40.58% | 23.86% | 34.66% | 47.06% |
| $Var(X_{LT})$ | 8.74% | 4.50% | 3.64% | 14.32% | 5.00% |
| $Cov(X_{ST}, X_{SN})$ | 2.91% | 2.53% | 2.20% | 2.00% | 2.17% |
| $Cov(X_{SN}, X_{LT})$ | −0.29% | 0.47% | 0.09% | 0.42% | 0.32% |
| $Cov(X_{ST}, X_{LT})$ | −0.01% | 0.00% | −0.01% | 0.00% | 0.01% |
| $Var(X_{LT}^{emis})$ | 2.49% | 4.88% | 1.80% | 4.97% | 1.90% |
| $Var(X_{LT}^{met})$ | 2.85% | 0.09% | 1.08% | 4.45% | 1.52% |
| $Cov(X_{LT}^{emis}, X_{LT}^{met})$ | 1.70% | −0.23% | 0.38% | 2.45% | 0.79% |

$^a$ Values of variances of each log-scale concentration time series
$Var(X) = Var(X_{ST}) + Var(X_{ST}) + Var(X_{ST}) + 2[Cov(X_{ST}, X_{SN}) + Cov(X_{SN}, X_{LT}) + Cov(X_{ST}, X_{LT})]$

**Table S2. Coefficients of determination ($R^2$) between each component and original time series ($X$) for Seoul average concentrations of PM$_{10}$, SO$_2$, NO$_2$, CO, and O$_{3\ 8h}$. Daily data for the period of July 2000 to Jun 2015 that $X_{LT}$ data is available were used.**

| Components | PM$_{10}$ | SO$_2$ | NO$_2$ | CO | O$_{3\ 8h}$ | Notes |
|---|---|---|---|---|---|---|
| $X_{ST}$ | 0.699 | 0.541 | 0.724 | 0.503 | 0.474 | Short-term components |
| $X_{SN}$ | 0.276 | 0.468 | 0.287 | 0.397 | 0.522 | Seasonal components |
| $X_{LT}$ | 0.081 | 0.055 | 0.038 | 0.152 | 0.057 | Long-term components |
| $X_{LT}^{emis}$ | 0.064 | 0.054 | 0.030 | 0.124 | 0.043 | Emission-related $X_{LT}$ |
| $X_{LT}^{met}$ | 0.069 | 0.003 | 0.019 | 0.107 | 0.040 | Meteorology-related $X_{LT}$ |

In Table 4, the sum of the linear trends of $X_{LT}^{emis}$ and $X_{LT}^{met}$ should be exactly the same as the $X_{LT}$ trend. However, the rounding after the calculation that we used here can induce tiny difference of ±1 in the last decimal place of both $\frac{\partial X_{LT}}{\partial t}$ and the sum of $\frac{\partial}{\partial t} X_{LT}^{emis}$ and $\frac{\partial}{\partial t} X_{LT}^{met}$. For example, $PM_{10LT}^{emis}$ trend ($-1.6883300\%$ yr$^{-1}$) and $PM_{10LT}^{met}$ trend ($-1.9456153\%$ yr$^{-1}$) exactly sum to $PM_{10LT}$ trend ($-3.6339453\%$ yr$^{-1}$). However, the trends rounded to the second decimal place are $-1.69\%$ yr$^{-1}$ for $PM_{10LT}^{emis}$ and $-1.95\%$ yr$^{-1}$ for $PM_{10LT}^{met}$, and thus their sum is $0.01\%$ yr$^{-1}$ larger than the rounded-off $PM_{10LT}$ trend ($-3.63\%$ yr$^{-1}$). For the result value, we think that the rounding after the calculation is more accurate than the rounding before the calculation. Therefore, we left Table 4 as the original version.

4. In eq. (9), should it be a residual term or $S_{LT}$ contains all terms except advection?

By Eq. (9), the term $S_{LT}$ should contain all effects on the change rate of long-term trend $(\frac{\partial X_{LT}}{\partial t})$, except the changes of inflow and outflow by horizontal advection $(-\vec{V}_{LT} \cdot \nabla X_{LT})$. The residual effects are source (surface emissions and atmospheric secondary production related to $X_{LT}^{emis(L)}$ and partly $X_{LT}^{met}$), sink (physical and chemical scavenging affected by $X_{LT}^{met}$), and diffusion (accumulation and dissipation), as well as the vertical advection term $(-w_{LT} \frac{\partial X_{LT}}{\partial z})$. The term $-w_{LT} \frac{\partial X_{LT}}{\partial z}$ is negligible because the long-term trend of vertical motion $(w_{LT})$ is close to zero. Therefore, $S_{LT}$ can be regarded as the sum of source term (directly related to $\frac{\partial}{\partial t} X_{LT}^{emis(L)}$) and sink and diffusion terms (affected by the long-term meteorological effect; $\frac{\partial}{\partial t} X_{LT}^{met}$).

5. What would be the main reason for CO(LT,Met) and SO$_2$(LT,Met) to show different trends though both of them are primary air pollutants?

The local meteorological factors such as wind speed, relative humidity, insolation, and temperature are not directly related to the regional transport of air pollutants, and thus can affect mainly on the contribution of local source pollutants rather than transported pollutants. In other words, if the local emissions at a local place are assumed to be zero, the local meteorological effects on accumulation and secondary production $(X_{LT}^{met})$ in the long-term air pollution trend $(X_{LT})$ will be negligible. As we wrote in the third paragraph on p.11, the emission intensity of CO in Seoul (215 t/km$^2$) is higher than those in the upwind source areas (e.g., 98 t/km$^2$ for the Chinese eastern coast), while that of SO$_2$ in Seoul (7 t/km$^2$) is much lower compared to those in the upwind source areas (e.g., 14 t/km$^2$ for the Chinese eastern coast). Therefore, the long-term SO$_2$ trend (SO$_{2LT}$) is much less contributed by the changes in meteorological effects on its local emission (SO$_{2LT}^{met}$) than by the changes in its regional emissions (SO$_{2LT}^{emis}$). As we describe in L9–13 on p.13, the interannual variation of satellite-based SO$_2$ level over China is similar to SO$_{2LT}$ (and SO$_{2LT}^{emis}$) in this study (Fig. 5c) and supports the large influence of the regional background SO$_2$ on the long-term SO$_2$ trend in Seoul.

6. The different trends of NO$_2$ and NO$_x$ might be caused not only by the reasons explained in the manuscript but also with changing oxidative potential of the atmosphere (for example, Kim and Lee, AAQR, in press).

We appreciate the reviewer's comment. L8–L12 on p.11 was now modified as follows:

[revised manuscript text omitted]
_{\text{LT}}(t) = \text{KZ}_{(365,3)}\varepsilon(t)[\varepsilon(t)] = X_{\text{BL}}(t) - X_{\text{SN}}(t)———$$

(4)

Then the seasonal component ($X_{\text{SN}}$), which represents the sum of the pure seasonal climatology ($X_{\text{BL}}^{\text{clm}}$) and the minor seasonality ($\varepsilon - \text{KZ}_{(365,3)}\varepsilon),[\varepsilon]),$ can be obtained by difference between $X_{\text{BL}}$ and $X_{\text{LT}}$.

5  Note that if we define $\chi_{\text{BL}} = \exp(X_{\text{BL}})$ and $\chi_{\text{ST}} = \exp(X_{\text{ST}})$ and employ the similar concept to the relationship between the original concentration and its log-transformation ($\chi = \exp(X)$), $\chi_{\text{BL}}$ represents the baseline concentration of the air pollutant, and $\chi_{\text{ST}}$ becomes the ratio of original concentration to baseline concentration ($\chi/\chi_{\text{BL}}$). Similarly, $\exp(X_{\text{SN}})$ and $\exp(X_{\text{LT}})$ can be defined as $\chi_{\text{SN}}$ and $\chi_{\text{LT}}$, respectively. Then $\chi_{\text{SN}}$ represents the seasonal change in concentration without trend, and $\chi_{\text{LT}}$ becomes the ratio of baseline concentration to detrended seasonal concentration ($\chi_{\text{BL}}/\chi_{\text{SN}}$).

10  As shown in the example with $PM_{10}$ in Seoul, the $\text{KZ}_{(15,5)}$ filter effectively removes $PM_{10_{\text{ST}}}$, of which period is smaller than 33 days, and leaves both the seasonality of high $PM_{10}$ concentrations in winter and spring and the long-term decreasing trend in $PM_{10_{\text{BL}}}$ (Figs. 2 and S4b). $PM_{10_{\text{SN}}}$ and $PM_{10_{\text{LT}}}$ well represent the seasonal variation, of which periods are between 33 days and 1.7 yr with representative periodicities of 0.5 yr and 1 yr, and the long-term variations, of which period are longer than 1.7 yr, respectively (Figs. 2 and S4c–d). The high levels in winter and spring in $PM_{10_{\text{SN}}}$ in Seoul is attributable to the shallow

15  boundary layer that traps local pollutants near the ground and frequent regional transport from China during the cold season (Kim et al., 2018).

**3.2 Separation of emission- and meteorology-related trends**

Since the long-term variability in air pollutant concentrations can be affected not only by changes in local and regional emissions but also by changes in meteorological conditions, the long-term trend is assumed to be consisted of meteorologically-

20  adjusted (emission-related) long-term component ($X_{\text{LT}}^{\text{emis}}$) and meteorology-related long-term component ($X_{\text{LT}}^{\text{met}}$). Therefore, the baseline can be represented as follows:

$$X_{\text{BL}}(t) = X_{\text{SN}}(t) + X_{\text{LT}}^{\text{met}}(t) + X_{\text{LT}}^{\text{emis}}(t) \tag{5}$$

To isolate the term $X_{\text{LT}}^{\text{emis}}$ in Eq. (5), we built a multiple linear regression model employing the baseline time series of the five representative meteorological variables ($\text{MET}_{\text{BL}}$), such as temperature ($T_{\text{BL}}$), sea level pressure ($P_{\text{BL}}$), relative humidity ($RH_{\text{BL}}$),

25  wind speed ($WS_{\text{BL}}$), and solar irradiance ($SI_{\text{BL}}$), which are obtained by the $\text{KZ}_{(15,5)}$ filter, as follows:

$$X_{\text{BL}}(t) = a_0 + \sum_i a_i \text{MET}_{\text{BL}_i}(t) + \varepsilon'(t),$$

$$\text{MET}_{\text{BL}} = [T_{\text{BL}}, P_{\text{BL}}, RH_{\text{BL}}, WS_{\text{BL}}, SI_{\text{
[revised manuscript text omitted]